# RESCUE: RETRIEVAL AUGMENTED SECURE CODE GENERATION

**Jiahao Shi**
Department of Computer Science
Purdue University
West Lafayette, IN, USA
shi768@purdue.edu

**Tianyi Zhang**
Department of Computer Science
Purdue University
West Lafayette, IN, USA
tianyi@purdue.edu

## ABSTRACT

Despite recent advances, Large Language Models (LLMs) still generate vulnerable code. Retrieval-Augmented Generation (RAG) has the potential to enhance LLMs for secure code generation by incorporating external security knowledge. However, the conventional RAG design struggles with the noise of raw security-related documents, and existing retrieval methods overlook the significant security semantics implicitly embedded in task descriptions. To address these issues, we propose RESCUE, a new RAG framework for secure code generation with two key innovations. First, we propose a hybrid knowledge base construction method that combines LLM-assisted cluster-then-summarize distillation with program slicing, producing both high-level security guidelines and concise, security-focused code examples. Second, we design a hierarchical multi-faceted retrieval that traverses the constructed knowledge base from top to bottom and integrates multiple security-critical facts at each hierarchical level, ensuring comprehensive and accurate retrieval. We evaluated RESCUE on four benchmarks and compared it with five state-of-the-art secure code generation methods on six LLMs. The results demonstrate that RESCUE improves the SecurePass@1 metric by an average of 4.8 points, establishing a new state-of-the-art performance for security. Furthermore, we performed in-depth analysis and ablation studies to rigorously validate the effectiveness of individual components in RESCUE. Our code is available at https://github.com/steven1518/RESCUE.

## 1 INTRODUCTION

Large language models (LLMs) have shown remarkable capabilities in coding-related tasks (Peng et al., 2023; Paradis et al., 2025). However, recent studies have revealed that LLMs often generate code with vulnerabilities (Pearce et al., 2022; Fu et al., 2023; Majdinasab et al., 2024). He & Vechev (2023); He et al. (2024) propose to finetune LLMs with security-aware objects. Yet these methods require significant effort in data curation and finetuning. Constrained decoding mechanisms can prevent the model from generating insecure code without finetuning (Li et al., 2024a; Fu et al., 2024). However, they require the availability of trained security models or human-crafted rules to serve as oracles or constraints to detect insecure code tokens during decoding. Another line of research (Nazzal et al., 2024; Kim et al., 2024) leverages security analysis tools such as Bandit (PyCQA, 2025) and SpotBugs (SpotBugs, 2025) to provide vulnerability feedback for iterative code refinement. However, these security analysis tools heavily rely on pre-defined static analysis logic and heuristics to find bugs and vulnerabilities, which makes them less flexible to incorporate new vulnerabilities and security knowledge. Furthermore, since these security analysis tools are often used to assess the security of LLM-generated code in evaluation (Nazzal et al., 2024; Kim et al., 2024), using them as a verifier to provide feedback in the code generation stage raises a data leakage concern.

Retrieval-Augmented Generation (RAG) offers a more flexible and training-free solution to incorporate security knowledge, such as documentation of secure coding practices and code examples. Despite some recent investigation (Zhang et al., 2024; Mukherjee & Hellendoorn, 2025; Tony et al., 2025), existing methods merely adapt conventional RAG methods to security domains and suffer from two limitations. First, security-related documents often contain information not relevant to the

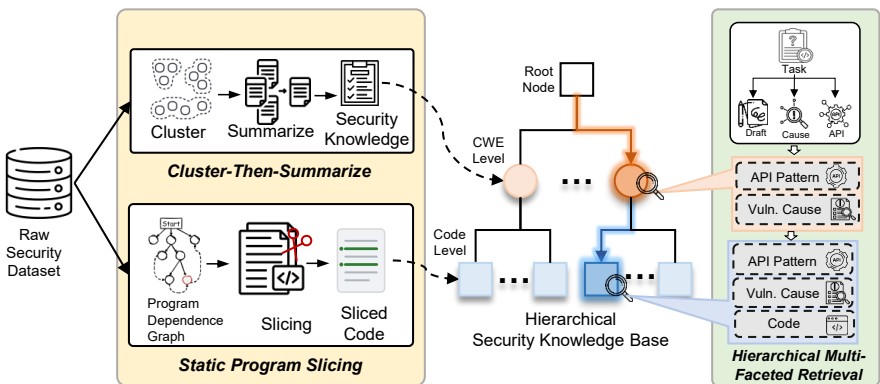

Figure 1: The overview framework of RESCUE.

target coding task, which would unnecessarily distract the LLM from generating code for the target task. For instance, a code example that demonstrates a secure coding practice may contain code logic of an example task that is very different from the target task. Second, the retrievers used by existing methods simply treat all security-related information as general text and measure similarity between texts as security relevance. They do not capture security semantics, such as the security requirements implicitly embedded in a task description. This oversight leads to inaccurate retrieval of security-related data.

To address these limitations, we propose **RESCUE**, a **RE**trieval-augmented **S**ecure **C**ode g**E**neration framework. RESCUE has two key innovations. First, we propose a hybrid knowledge base construction method that combines semantic summarization with static program analysis. Given raw security data, an LLM-assisted cluster-then-summarize pipeline first distills generalizable security knowledge as *high-level guidelines* (e.g.,"*replace* `yaml.load()` *with* `yaml.safe_load()` *to prevent code execution vulnerabilities*"). Then, a static program slicing procedure extracts *concise, security-focused code examples* by isolating the statements relevant to vulnerability fixes while filtering out unrelated logic. Second, we develop a hierarchical multi-faceted retrieval method that performs a coarse-grained search to identify relevant vulnerability types and secure code examples. In each search iteration, our method proactively analyzes three security-critical aspects—API pattern, vulnerability cause analysis, and code—and then fuses these separated faceted search results to obtain precise security knowledge to guide code generation.

We evaluate RESCUE using five baseline methods and six LLMs across four benchmarks. The results demonstrate that on average, RESCUE achieves an absolute improvement of 4.8% in SecurePass@1, establishing a new state-of-the-art for security. Meanwhile, RESCUE retains 98.7% of the original models' capability of generating functionally correct code. Furthermore, we have conducted ablation studies to confirm the contributions of both the security knowledge base construction and the proposed hierarchical multi-faceted retrieval method to the enhanced performance of RESCUE.

In summary, our contributions are threefold: (1) We propose a novel hybrid distillation method that synergistically combines LLM-based summarization with program slicing to construct a hierarchical security knowledge base. (2) We design a security-aware, hierarchical multi-faceted retrieval strategy that improves relevance by analyzing and fusing multiple security-critical aspects of a task. (3) We perform a comprehensive evaluation that not only establishes the state-of-the-art performance on four benchmarks but also provides in-depth analyses that validate our design choices.

## 2 METHOD

Figure 1 illustrates the overview of RESCUE, which operates in two core stages. In the offline stage, we construct a hierarchical security knowledge base from raw security data. In the online stage, we design a hierarchical multi-faceted retrieval method to query relevant security knowledge for a given coding task. The retrieved security knowledge forms a tailored *security context* that augments the prompt of task to steer the LLM toward generating secure code.

## 2.1 HIERARCHICAL SECURITY KNOWLEDGE BASE CONSTRUCTION

Inspired by how human experts reason about security problems using taxonomies (Igure & Williams, 2008), we propose modeling security knowledge as a hierarchical structure, with generalizable knowledge at the high level and concise code examples at the low level. In particular, we leverage the Common Weakness Enumeration (CWE) (MITRE, 2025) to categorize vulnerabilities and construct this hierarchy.

In this work, we focus on constructing a security knowledge base from historical vulnerabilities and their fixes. Figure 2 shows an example. Such security data are widely available in many security databases (CVE Program, 2025; National Institute of Standards and Technology (NIST), 2024; GitHub, 2024). Specifically, we use the training dataset from SafeCoder (He et al., 2024) as our raw security dataset in this work. This dataset includes 704 vulnerabilities and their fixes from CVE and GitHub projects. Please refer to Appendix A for more details.

Given the distinct nature of knowledge at different levels, we introduce a hybrid method for constructing the security knowledge base that combines semantic summarization with static program analysis. At the

```
--- vulnerability.py
+++ fixing.py
 def puppet_enc_default():
  ...
  # Get the default YAML
  curd = db.cursor()
  curd.execute("SELECT value FROM
      kv_settings WHERE key = 'puppet.enc
      .default'")
  result = curd.fetchone()
  classes = result['value']
  ...
  # Validate classes YAML
- data = yaml.load(classes)
+ data = yaml.safe_load(classes)
  ...
```

Figure 2: A known vulnerability and its fix that demonstrates the replacement of unsafe `yaml.load()` with `yaml.safe_load()`, where red-highlighted code shows vulnerable lines removed and green-highlighted code indicates security fixes added (See Appendix E for the complete example).

high level, an LLM-assisted *cluster-then-summarize* pipeline distills generalizable security knowledge. At the low level, a *static program slicing* procedure extracts concise, security-focused code examples by isolating the statements relevant to vulnerability fixes while filtering out unrelated logic.

### 2.1.1 CLUSTER-THEN-SUMMARIZE

Raw vulnerability-fix instances are often verbose and instance-specific, making them unsuitable as direct general guidance. Although CWE has provided concise descriptions, they lack actionable fixing strategies and are too abstract to be effective for secure code generation. Prior work (He et al., 2024) demonstrates that using CWE descriptions in context does not significantly improve the security of LLM-generated code. To address this gap, we introduce a *cluster-then-summarize* pipeline to distill generalizable security knowledge from raw vulnerability-fix instances.

First, we group the raw data instances into clusters based on their associated CWE type. Then, for each cluster, we apply a bottom-up summarization process using an LLM. This process begins by summarizing small, fixed-size subsets of instances. Next, it recursively summarizes the outputs from the previous results until a single, cohesive summary is generated for the entire cluster. This recursive, bottom-up summarization approach allows us to effectively process a large volume of data while maintaining high-quality, comprehensive outputs. Details of the algorithm are provided in Appendix B.1.

Since this preprocessing step is a one-time effort, we adopt GPT-4o in an offline setting. Ultimately, this pipeline generates two summaries for each CWE category. First, it produces **security guidelines** that define actionable instructions and best practices for preventing specific vulnerabilities as *high-level knowledge*. Second, it summarizes **vulnerability causes**, which capture the root conditions and failure patterns that lead to the occurrence of the vulnerability. These distilled causes are subsequently leveraged in the retrieval stage as a key security aspect.

### 2.1.2 SECURITY-FOCUSED STATIC PROGRAM SLICING

As shown in Figure 2 and Appendix E, security code examples may contain code logic (e.g., database connection, web scraping) that is not relevant to a target programming task at inference time. Such code logic may distract the LLM from generating functional code aligned with the target task. Therefore, we design a security-focused slicing method to extract concise code examples with only security-related program statements from raw code examples.

RESCUE begins by building a Program Dependence Graph (PDG) that captures how statements in the code depend on each other through both data and control dependence. Based on the statements influenced by patches, it identifies points of interest by treating deleted statements as indicators of vulnerable code and added statements as indicators of secure code. Around these points, we perform bidirectional slicing to extract the relevant context: backward slicing traces the statements that influence the vulnerable or secure code, while forward slicing captures those that are affected by it. Finally, the sliced subgraphs from the vulnerable and secure versions are compared and complemented with missing context, resulting in two parallel, contextually complete code variants. Appendix B.2 describes this security-focused slicing algorithm in detail.

## 2.2 Hierarchical Multi-Faceted Retrieval

With the knowledge base constructed, the next step is to retrieve the most relevant security knowledge for a given task. Our design mirrors the hierarchy of knowledge base: retrieval begins at the CWE level to identify potential relevant vulnerability types, and then narrow down to fine-grained secure code examples. A key novelty of our method is the *multi-faceted retrieval strategy*, which conducts proactive analysis to generate multiple security-aware queries. We will first explain this multi-faceted design and then describe the hierarchical retrieval process.

### 2.2.1 Proactive Multi-Faceted Analysis

Conventional retrieval typically relies on task functional descriptions, which lack explicit security semantics. In contrast, our method proactively analyzes coding tasks from multiple security-critical perspectives. This proactive stance is essential because security vulnerabilities often emerge from subtle implementation details that are not apparent in task descriptions. Specifically, we consider the following three facets:

(1) **Vulnerability Cause Analysis:** To explicitly clarify the security requirements and potential attacks, we instruct LLMs to analyze the task and explain the underlying vulnerability cause $V_{\text{cause}}$ (see prompt in Appendix D.2.2).
(2) **Draft Code Generation:** Since vulnerabilities often manifest during the coding process, we generate an initial code $C_{\text{draft}}$ using zero-shot prompting (details in Appendix D.2.1).
(3) **API Call Extraction:** Since security vulnerabilities frequently stem from API misuse (Zhang et al., 2018; Li et al., 2021; Egele et al., 2013), we apply the visitor pattern to traverse the abstract syntax trees of the draft code $C_{\text{draft}}$ to identify all API calls.

### 2.2.2 Hierarchical Retrieval Process.

Building on these proactive multi-faceted analyses, RESCUE begins a two-step retrieval process.

*Step 1: CWE-level Retrieval.* RESCUE first selects the top-k relevant CWE types using two facets: (1) **Vulnerability Cause Analysis**: RESCUE uses a widely used dense retriever for RAG, *bge-base-en-v1.5* (Xiao et al., 2023), to compute $\text{score}_{\text{VCA}}$ between input task's vulnerability cause and the indexed vulnerability causes per CWE type. These per-CWE vulnerability causes are distilled from raw instances during the offline stage (Section 2.1.1). (2) **API Pattern**: Since API calls are discrete, RESCUE computes $\text{score}_{\text{API}}$ between the draft code's APIs and the API set of each CWE type via sparse retrieval, BM25 (Robertson et al., 1995). Each CWE's API set is constructed by aggregating all APIs extracted from its associated vulnerability-fix instances.

Then, it fuses these two facet scores using a modified Reciprocal Rank Fusion (RRF) method with thresholding and rank-based filtering. This approach is motivated by the observation that only certain scenarios require security-specific guidance. Our modified RRF is defined as:

$$\text{RRF}(d) = \sum_{i=1}^{f} V_i(d) \cdot \frac{1}{r_i(d) + \alpha}, \quad \text{where } V_i(d) = \mathbb{I}(s_i(d) > \tau_i) \cdot \mathbb{I}(r_i(d) \leq 10). \quad (1)$$

Here, $s_i(d)$ and $r_i(d)$ are the score and rank of item $d$ for facet $i$, $\tau_i$ is a confidence threshold, and $\alpha$ is a smoothing parameter.

*Step 2: Code-level Retrieval.* After narrowing down to relevant CWE types, we proceed to conduct a fine-grained search at the low level using the same two facets and an additional third facet, **Code**

**Similarity**. We also employ dense retrieval to obtain $score_C$ between the draft code and sliced secure code examples, capturing similarity at the code level to identify relevant secure patterns.

The scores from all three facets are fused via the same modified RRF to select the most relevant secure code examples. Finally, we use the security guidelines corresponding to the selected example's CWE type, along with the sliced secure code example, to construct prompts for LLMs (detailed in Appendix D.3), guiding secure code generation.

# 3 EXPERIMENTS

## 3.1 EXPERIMENT SETUP

**Models** We evaluate RESCUE with six LLMs from different model families and with different model sizes, including GPT-4o-mini (OpenAI, 2024), Llama3.1-8B-Instruct (Dubey et al., 2024), Qwen2.5-Coder-7B-Instruct, Qwen2.5-Coder-32B-Instruct (Hui et al., 2024), DeepSeek-Coder-V2-Lite-Instruct (Zhu et al., 2024), and DeepSeek-V3-0324 (Liu et al., 2024). For inference, GPT-4o-mini and DeepSeek-V3 are accessed via official APIs, while the other models are locally deployed using vLLM (Kwon et al., 2023) on an NVIDIA A800 80GB GPU.

**Benchmarks** We first evaluate RESCUE on a state-of-the-art security benchmark called Code-Guard+ (Fu et al., 2024). CodeGuard+ incorporates and extends prior security benchmarks (Pearce et al., 2022; Siddiq & Santos, 2022). It includes 94 security-sensitive coding scenarios, each of which is accompanied by unit tests and CodeQL checks, enabling the testing of both functional correctness and security. CodeQL (GitHub, 2025) is a widely used security analysis tool and has been used to evaluate the security of LLM-generated code in several studies (He & Vechev, 2023; He et al., 2024; Li et al., 2024a).

Furthermore, following prior work (He et al., 2024; Li et al., 2024a; Zhang et al., 2024), we also use regular code generation benchmarks to demonstrate that RESCUE does not affect the functional correctness of LLM-generated code while improving its security. Specifically, we evaluate RESCUE on HumanEval+ (Liu et al., 2023a), BigCodeBench (Zhuo et al., 2025), and LiveCodeBench (Jain et al., 2025). HumanEval+ is a popular code generation benchmark with 164 basic programming tasks and extensive test cases. BigCodeBench is a much bigger and more challenging benchmark. It includes 1140 programming tasks that span across various scenarios, such as data analysis and web development, and involve complex function calls. LiveCodeBench is specifically designed to address the data contamination problem in LLM code generation. It includes a continuously updated set of code generation problems. We use its `release-v5` version, collected by January 2025, which has 880 code generation problems.

To reduce the randomness of code generation, we generate 100 samples for each problem on CodeGuard+ and HumanEval+. Since BigCodeBench and LiveCodeBench have roughly 10× more problems than CodeGuard+ and HumanEval+, we generate 10 samples on these two benchmarks.

**Metrics** Following prior work (Zhang et al., 2024; He et al., 2024; Li et al., 2024a; He & Vechev, 2023), we used SecureRate to measure the security of LLM-generated code. SecureRate is defined as the proportion of generated code samples that pass the security checks, e.g., CodeQL checks in CodeGuard+ (Fu et al., 2024). For functional correctness measurement, we followed recent work (Zhuo et al., 2025; Jain et al., 2025) to use the unbiased Pass@$k$ (Chen et al., 2021).

In the meantime, we observe that SecureRate overlooks the functional correctness of the generated code. For instance, a code snippet that does nothing will always be secure but is meaningless in practice. As shown in Table 1, several methods excessively sacrifice functionality to achieve higher SecureRate scores, which fails to reflect genuine security improvements. Therefore, we introduce a new metric for security evaluation, SecurePass@$k$, which jointly evaluates functionality and security:

$$\text{SecurePass@}k := \underset{\text{Problems}}{\mathbb{E}} \left[ 1 - \frac{\binom{n-sp}{k}}{\binom{n}{k}} \right] \qquad (2)$$

where $n$ is the total number of generated samples, $k$ represents the number of our observed samples, and $sp$ means the number of samples that pass both unit tests and CodeQL security checks.

Specifically, we report SecureRate, Pass@1, and SecurePass@1 on the CodeGuard+ benchmark, since it includes both security checks and unit tests. For HumanEval+, BigCodeBench, and LiveCodeBench, we only report Pass@1 since they only include unit tests to check functional correctness.

**Baseline Methods** We compare our approach with five state-of-the-art baseline methods, each representing a different strategy for secure code generation: (1) **SecCoder** (Zhang et al., 2024) is a RAG method that retrieves the most similar secure code example to facilitate in-context learning. To ensure a fair comparison, we utilize the same retriever *bge-base-en-v1.5* (Xiao et al., 2023) and the raw security data introduced in Section 2 as retrieval documents. (2) **Codexity** (Kim et al., 2024) utilizes an external security tool to identify vulnerabilities for generated code and then iteratively refines the code based on detection results, up to three iterations. To avoid data leakage, we use another widely adopted security tool, semgrep (Semgrep, 2025), instead of CodeQL. (3) **SafeCoder** (He et al., 2024) adopts a specialized instruction-tuning method for code security. We use the training dataset and hyperparameters from the original paper to finetune the LLMs used in our experiments. Given the limited computational resources, we only fine-tune Qwen2.5-Coder-7B-Instruct, Llama3.1-8B-Instruct, and Deepseek-Coder-V2-Lite-Instruct, excluding larger models and closed-source models. (4) **CoSec** (Li et al., 2024a) manipulates the decoding process by controlling the output token logits based on a small trained security-specialized model. For implementation, we use the same training dataset as in the original paper. Since this method requires training a smaller model that uses the same tokenizer to perform its decoding, we select Qwen2.5-Coder-0.5B-Instruct as the small security-specialized model for Qwen2.5-Coder-7B-Instruct and 32B and Llama3.1-1B-Instruct for Llama3.1-8B-Instruct. Other models are either closed-source or lack a smaller version from the same model family. (5) **INDICT** (Le et al., 2024) is a multi-agent debate framework with external security tools to generate both security and helpfulness critiques for iterative code refinement. Following the original setup, we set the iteration round to three and use all four tools for code generation.

**Other Implementation Details** We use `tree-sitter` (Brunsfeld & Github) to implement static program analysis, slicing, and API extraction. To control the length of sliced code, our method performs 2-hop program slicing. During generation, we set the temperature to 0.2. To balance the precision and recall, we use a top-k value of 4. To only search security knowledge for security-relevant problems, the thresholds for the facets of API, vulnerability cause, and code are set to 4.0, 0.75, and 0.65, respectively. We follow prior work (Cormack et al., 2009) to set the RRF parameter $\alpha$ to 60.

## 3.2 MAIN RESULTS

As shown in Table 1, RESCUE consistently outperforms all existing methods in terms of Se-curePass@1, the metric that balances security and functional correctness of generated code. Specifically, RESCUE achieves 4.8% absolute improvement on average compared with the second-best baseline. We also observed that several existing approaches sacrifice functional correctness for security. For example, though INDICT (Le et al., 2024) achieves a higher SecureRate than RESCUE, it has the lowest Pass@1 among all methods. Furthermore, when evaluated on the three benchmarks on functional correctness, RESCUE achieves comparable performance compared with the original models. This indicates that RESCUE does not severely damage the functional correctness of generated code while improving its security. Appendix C.1 provides a more direct comparison of the performance improvements achieved by each method.

In contrast, another RAG-based method, SecCoder (Zhang et al., 2024), does not significantly improve SecurePass@1 or SecureRate. This suggests that simply applying conventional RAG for secure code generation cannot fully exert the security knowledge. Section 3.3 shows the ablation study results of the hierarchical knowledge base and retrieval method in RESCUE.

## 3.3 IN-DEPTH ANALYSIS RESULTS

To thoroughly analyze each component of RESCUE, we conducted extensive experiments on the CodeGuard+ benchmark using three models: Deepseek-Coder-V2-Lite, Qwen2.5-Coder-7B, and Llama3.1-8B. We selected these models because they represent three widely adopted model families and have demonstrated strong performance when integrated with our proposed method.

Table 1: Performance comparison across six LLMs and four benchmarks: CodeGuard+, HumanEval+ (HE+), BigCodeBench (BCB), and LiveCodeBench (LCB). For each model, **bold** indicates the best score, and underline indicates the second-best result. Notably, SafeCoder requires fine-tuning the LLMs and CoSec requires training a smaller model with the same tokenizer, therefore, we evaluate both methods on applicable open-source LLMs.

| Model | Method | CodeGuard+ | | | HE+ | BCB | LCB |
|---|---|---|---|---|---|---|---|
| | | SP@1 | SR | Pass@1 | Pass@1 | Pass@1 | Pass@1 |
| **DeepSeek-Coder-V2-Lite** | LLM alone | 59.7 | 65.1 | 86.8 | **73.0** | 39.5 | 24.4 |
| | SecCoder | 55.6 | 63.7 | 82.9 | 64.7 | **40.9** | 23.9 |
| | Codexity | 58.4 | 70.3 | 80.1 | 70.4 | 35.2 | 24.4 |
| | INDICT | 49.5 | 72.3 | 62.5 | 57.9 | 32.4 | 23.4 |
| | SafeCoder | 60.3 | 68.6 | 81.7 | 61.2 | 14.8 | 13.4 |
| | RESCUE | **65.6** | **72.8** | **87.9** | 70.4 | 39.1 | **25.2** |
| **Qwen2.5-Coder-7B** | LLM alone | 51.2 | 61.3 | 83.3 | 77.1 | **45.2** | 24.5 |
| | SecCoder | 54.9 | 65.6 | 79.8 | 74.6 | 42.3 | 24.2 |
| | Codexity | 51.4 | 68.1 | 77.7 | 75.1 | 44.9 | **24.8** |
| | INDICT | 41.9 | **84.1** | 48.5 | 71.3 | 33.8 | 22.4 |
| | CoSec | 52.8 | 64.6 | 82.8 | 68.0 | 19.0 | 23.3 |
| | SafeCoder | 56.5 | 74.0 | 75.1 | 67.7 | 43.0 | 19.6 |
| | RESCUE | **64.8** | 72.1 | **86.2** | **77.8** | 42.9 | 24.3 |
| **Qwen2.5-Coder-32B** | LLM alone | 59.3 | 66.0 | **88.3** | 80.0 | **54.1** | 22.3 |
| | SecCoder | 58.1 | 66.6 | 84.7 | **82.9** | 53.9 | 25.2 |
| | Codexity | 57.1 | 71.6 | 80.7 | 79.5 | 53.4 | 22.0 |
| | INDICT | 40.4 | **84.4** | 48.1 | 65.9 | 33.8 | 20.9 |
| | CoSec | 41.9 | 54.9 | 65.1 | 78.0 | 52.7 | 23.4 |
| | RESCUE | **65.1** | 81.5 | 80.6 | 80.8 | 49.9 | **27.1** |
| **Llama3.1-8B** | LLM alone | 53.7 | 63.7 | **82.8** | 58.8 | 36.0 | **15.0** |
| | SecCoder | 48.4 | 60.6 | 75.4 | 51.0 | 35.6 | 14.9 |
| | Codexity | 50.8 | 68.5 | 73.7 | 57.7 | **37.3** | 14.7 |
| | INDICT | 16.0 | **79.8** | 19.8 | 15.2 | 8.8 | 7.4 |
| | CoSec | 53.9 | 62.2 | 78.3 | 55.7 | 4.9 | 5.8 |
| | SafeCoder | 52.4 | 69.4 | 72.4 | 54.0 | 31.4 | 11.0 |
| | RESCUE | **56.2** | 69.7 | 77.6 | 54.6 | 31.8 | 14.7 |
| **GPT-4o-mini** | LLM alone | 58.2 | 68.9 | **80.7** | **83.6** | **54.6** | 37.6 |
| | SecCoder | 57.8 | 71.2 | 79.7 | 82.6 | 54.0 | 37.4 |
| | Codexity | 55.5 | 77.0 | 72.5 | 83.3 | 52.6 | **38.1** |
| | INDICT | 31.1 | **85.5** | 36.8 | 51.2 | 26.0 | 35.7 |
| | RESCUE | **63.0** | 77.6 | 77.3 | 81.3 | 45.2 | 37.5 |
| **DeepSeek-V3-0324** | LLM alone | 64.6 | 71.4 | **87.4** | 72.9 | 61.5 | **64.3** |
| | SecCoder | 64.7 | 74.8 | 83.4 | 79.6 | **61.7** | 63.6 |
| | Codexity | 63.5 | 77.4 | 80.0 | 74.3 | 60.9 | 64.1 |
| | INDICT | 29.6 | **85.6** | 32.4 | 68.3 | 31.9 | 63.6 |
| | RESCUE | **69.7** | 79.1 | 83.5 | **89.1** | 60.2 | **64.3** |

Table 2: Results of ablation studies on security knowledge base construction, evaluating five variants of the security knowledge base with three models on the CodeGuard+ benchmark across four metrics: number of input tokens (#Token), SecurePass@1 (SP@1), SecureRate (SR), and Pass@1 (P@1). The **bold** number indicates the best performance, and "—" represents dismissal.

| Setting | DSC-V2-Lite-Instruct | | | | Qwen2.5-Coder-7B-Instruct | | | | Llama3.1-8B-Instruct | | | |
|---|---|---|---|---|---|---|---|---|---|---|---|---|
| | #Token↓ | SP@1 | SR | P@1 | #Token↓ | SP@1 | SR | P@1 | #Token↓ | SP@1 | SR | P@1 |
| RESCUE | 503 | **65.6** | 72.8 | **87.9** | 397 | 64.8 | 72.1 | 86.2 | 428 | **56.2** | 69.7 | **77.6** |
| w/o construction | — | 55.6 | 61.9 | 81.0 | — | 53.9 | 63.6 | 75.4 | — | 45.3 | 58.8 | 73.3 |
| w/o guideline | — | 63.3 | 72.1 | 86.2 | — | 63.9 | **72.2** | 86.4 | — | 51.4 | 65.3 | 76.9 |
| w/o slicing | 661 | 61.0 | 70.5 | 80.6 | 506 | 62.6 | 71.3 | 83.1 | 610 | 52.7 | 68.8 | 74.3 |
| w/o ps$_{generation}$ | 753 | 64.8 | 71.9 | 86.5 | 595 | **66.1** | 71.3 | **87.6** | 698 | 53.5 | 67.0 | 75.4 |
| w/o ps$_{retrieval}$ | **435** | 63.1 | **73.1** | 83.4 | **358** | 64.8 | 71.4 | 86.6 | **359** | 54.0 | **71.3** | 76.5 |

### 3.3.1 ABLATION STUDY ON SECURITY KNOWLEDGE BASE CONSTRUCTION

To better understand the contributions of the security knowledge base construction components, specifically security guideline extraction and program slicing, we performed ablation studies using five variants: (1) *w/o construction*: Does not utilize the constructed security knowledge base and

instead directly employs the raw security data. (2) *w/o guideline*: Removes the security guidelines from the constructed security knowledge base, providing only relevant secure code examples to the model. Since sliced code examples are used in both the retrieval and generation stages, we designed three additional detailed variants for further investigation: (3) *w/o slicing*: Completely removes the slicing component, using original (unsliced) code in both retrieval and generation stages. (4) *w/o $ps_{retrieval}$*: Replaces sliced code with original code only during the retrieval stage. (5) *w/o $ps_{generation}$*: Replaces sliced code with original code only during the generation stage. Table 2 shows the experimental results, from which we derive the following key observations:

**Raw security data alone does not improve performance; constructing a security knowledge base is essential.** Comparing RESCUE with the *w/o construction* variant, we observe a significant performance drop in all metrics when using raw security data directly. This is because irrelevant code logic distracts the models during code generation, and the noise of irrelevance also leads to inaccurate retrieval results. Thus, constructing a refined security knowledge base is crucial.

**Security guidelines enhance security performance.** The results of the *w/o guideline* variant show a decrease in SP@1 compared to RESCUE. For instance, the SP@1 metric in Llama3.1-8B-Instruct drops by 4.8 points, highlighting the substantial contribution of security guidelines.

**Program slicing improves security performance and reduces token costs.** The results of the *w/o slicing* variant demonstrate the effectiveness of program slicing in enhancing performance. Additionally, finer-grained ablation results from the *w/o $ps_{retrieval}$* and *w/o $ps_{generation}$* variants reveal that program slicing is beneficial in both the retrieval and generation stages. Specifically, slicing helps retrieve more relevant security knowledge and provides concise yet informative code examples during generation. Finally, we note that applying sliced code during generation substantially reduces the number of input tokens. Additional analysis comparing lines of code before and after program slicing is provided in the Appendix C.2.

### 3.3.2 IMPACT ANALYSIS OF HIERARCHICAL RETRIEVAL

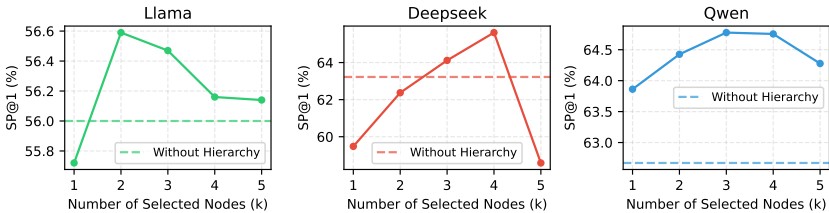

Figure 3: Comparison of SecurePass@1(SP@1) performance across hierarchical and non-hierarchical settings for three LLMs, DeepSeek-Coder-V2-Lite (Deepseek), Llama3.1-8B-Instruct (Llama), Qwen2.5-Coder-7B (Qwen), at varying numbers of selected CWE types (k).

To analyze the effectiveness of our hierarchical design, we conducted an impact analysis by varying the the parameter $k$ in top-k relevant CWE types from 1 to 5 and comparing against non-hierarchical baselines, as shown in Figure 3. The peak results of hierarchy consistently outperform the non-hierarchical setting across all LLMs. Interestingly, the performance exhibits an inverted U-shaped trend as $k$ increases: smaller $k$ values may restrict the model to local optima, while larger $k$ values introduce noise, diminishing performance. Based on these insights, we select $k = 4$ as the optimal configuration for all experiments.

### 3.3.3 ABLATION STUDY ON MULTI-FACETED RETRIEVAL

To systematically evaluate the contribution of each facet to retrieval performance, we conducted an ablation study covering all seven possible combinations of the three facets under consideration. We make two key observations based on the results in Table 3:

**Multi-faceted retrieval mostly outperforms single-facet approaches.** Overall, single-facet retrieval methods show limited and inconsistent effectiveness. In contrast, our multi-faceted retrieval effectively leverages complementary strengths from individual facets, outperforming each single-facet method.

Table 3: Results of ablation studies on multi-faceted retrieval, evaluating seven combinations of API pattern (API), vulnerability cause analysis (VA), and code similarity (Code) across three different LLMs on the CodeGuard+ benchmark. In the table, "✓" indicates the adoption of a facet, while "—" represents its dismissal. The **bold** number indicates the best performance.

| Facet | | | DeepSeek-Coder-V2-Lite | | | Llama3.1-8B | | | Qwen2.5-Coder-7B | | |
|---|---|---|---|---|---|---|---|---|---|---|---|
| API | VA | Code | SP@1 | SecurityRate | Pass@1 | SP@1 | SecurityRate | Pass@1 | SP@1 | SecurityRate | Pass@1 |
| ✓ | ✓ | ✓ | **65.6** | 72.8 | **87.9** | **56.2** | 69.7 | 77.6 | 64.8 | 72.1 | **86.2** |
| ✓ | — | ✓ | 62.7 | 71.5 | 84.8 | 53.4 | 67.7 | 77.0 | 63.8 | 70.6 | **86.2** |
| ✓ | ✓ | — | 64.1 | **73.5** | 84.5 | 54.8 | 68.0 | 77.0 | 64.6 | 71.5 | 86.1 |
| — | ✓ | ✓ | 61.3 | 72.4 | 83.0 | 51.7 | 68.4 | 75.4 | 65.0 | 72.3 | 85.3 |
| ✓ | — | — | 61.4 | 71.7 | 82.9 | 52.1 | 67.4 | 76.6 | 62.2 | 69.9 | 84.9 |
| — | ✓ | — | 63.2 | 71.9 | 86.2 | 53.6 | **70.0** | 73.2 | **66.2** | **72.5** | **86.2** |
| — | — | ✓ | 57.1 | 68.3 | 82.1 | 51.1 | 62.5 | **81.2** | 61.7 | 69.1 | 84.8 |

Notably, combining all three facets mostly achieves the best overall performance across nearly all scenarios, highlighting the advantage of integrating diverse security-related facets.

**Our proposed API pattern and vulnerability cause analysis facets significantly outperform the code similarity facet.** The code-only facet consistently lags behind other facet combinations, reinforcing that dense retrieval approaches based solely on code similarity are insufficient. Incorporating API pattern and vulnerability cause analysis facets substantially enhances retrieval accuracy, demonstrating their effectiveness in capturing meaningful semantic context.

## 3.4 IMPACT ANALYSIS OF SUMMARIZATION LLMs

Since our pipeline employs GPT-4o for the summarization step, one potential concern is whether the observed improvements stem merely from relying on a powerful black-box model. To address this, we conduct an impact analysis by replacing GPT-4o with three open-source summarization models: Qwen2.5-Coder-7B, Llama3.1-8B, and DeepSeek-V3. In this setting, the *summarization model* is used to distill security knowledge, while the *generation model* is applied during the online stage for secure code generation. As shown in Table 4, **the performance gains do not depend on GPT-4o; they consistently arise from our pipeline design. Even with smaller open-source models, the framework achieves comparable improvements.**

Table 4: Impact of different summarization and generation model combinations. The improvements hold across models, showing that gains stem from the pipeline rather than reliance on GPT-4o.

| Summarization Model | Generation Model | SP@1 | SR | P@1 |
|---|---|---|---|---|
| Qwen2.5-Coder-7B | Qwen2.5-Coder-7B | 60.7 | 72.0 | 83.3 |
| | Llama3.1-8B | 56.9 | 69.2 | 78.9 |
| | DeepSeek-V3 | 69.9 | 80.0 | 83.0 |
| Llama3.1-8B | Qwen2.5-Coder-7B | 55.6 | 69.5 | 79.8 |
| | Llama3.1-8B | 51.2 | 66.7 | 75.9 |
| | DeepSeek-V3 | 71.5 | 79.2 | 85.2 |
| DeepSeek-V3 | Qwen2.5-Coder-7B | 59.2 | 71.3 | 80.3 |
| | Llama3.1-8B | 56.5 | 72.1 | 77.7 |
| | DeepSeek-V3 | 68.6 | 79.5 | 82.7 |

## 3.5 ADDITIONAL ANALYSIS

Beyond the main experiments, we conduct a series of additional analyses to provide a more comprehensive evaluation of RESCUE. First, we analyze the **computational overhead** of RESCUE, showing that the additional latency is acceptable and can be further reduced through concurrent LLM calls (Appendix C.3). Second, we evaluate RESCUE on **CWEval** (Peng et al., 2025), a dynamic testing benchmark, demonstrating that our gains are not artifacts of static analysis tools (Appendix C.4). Third, we examine the **generalizability** of RESCUE from multiple perspectives: we analyze its performance on both seen and unseen CWE types (Appendix C.5.2), and further evaluate it on programming languages absent from our knowledge base, confirming that the distilled security knowledge transfers broadly across languages (Appendix C.5.3). Fourth, we validate our modified RRF fusion strategy through ablation and sensitivity analysis of threshold hyperparameters (Appendix C.6). Finally, we break down performance by vulnerability category, revealing that RESCUE yields stronger gains on non-functionality-specific vulnerabilities due to their more explicit and localizable constraints (Appendix C.7).

## 4 RELATED WORK

**Retrieval-Augmented Code Generation** Recent research has investigated RAG to enhance code generation (Yang et al., 2025; Lu et al., 2022; Gao et al., 2024; Tan et al., 2025). Several studies focus on repository-level retrieval for code generation (Wu et al., 2024; Zhang et al., 2023). Others introduce external API documentation to aid generation involving unfamiliar APIs (Zan et al., 2022; Zhou et al., 2023; Liu et al., 2023b; Gu et al., 2025). Additionally, retrieval of functionally similar examples has been used to enhance functional correctness (Parvez et al., 2021; Su et al., 2024; Nashid et al., 2023). In contrast, our method specifically targets retrieval of security knowledge to improve the security of generated code without compromising functional correctness.

**Secure Code Generation** Existing studies have identified significant security concerns in LLM-generated code (Hou et al., 2024; GitHub, 2024; Cursor, 2024). To mitigate these, recent approaches include fine-tuning models on security-specific datasets or tasks (He & Vechev, 2023; He et al., 2024; Hajipour et al., 2024; Li et al., 2024b; Xu et al., 2025), and training-free approaches such as prompt engineering (Tony et al., 2024), security analysis tool integration (Kim et al., 2024; Nazzal et al., 2024), agent (Le et al., 2024), and RAG frameworks (Mukherjee & Hellendoorn, 2025; Zhang et al., 2024; Tony et al., 2025; Lin et al., 2025). Specifically, SecCoder (Zhang et al., 2024) retrieves secure code examples with dense retriever and SOSecure (Mukherjee & Hellendoorn, 2025) retrieves StackOverflow content with BM25. Our work automatically constructs a hierarchical security knowledge base from raw security data and proposes a specially designed retrieval method.

## 5 CONCLUSION

This work introduces RESCUE, a novel retrieval-augmented secure code generation framework that adopts a hybrid distillation method to construct a hierarchical security knowledge base and designs a hierarchical multi-faceted retrieval method. Compared to five state-of-the-art methods across four benchmarks and six models, RESCUE demonstrates substantial improvements in security without compromising functional correctness. Further in-depth analyses highlight the necessity of knowledge base construction and validate the effectiveness of our proposed hybrid distillation method. Additional thorough analyses confirm the advantages of our hierarchical multi-faceted retrieval design.

## ACKNOWLEDGMENT

We sincerely thank the anonymous reviewers for their constructive feedback. This work was supported in part by the National Science Foundation (NSF) under Awards NSF CAREER 2340408 and NSF Proto-OKN 2333736.

## ETHICS STATEMENT

Our work adheres to the ICLR Code of Ethics. The primary goal of this research is to enhance the security of code generated by Large Language Models (LLMs), thereby reducing the prevalence of software vulnerabilities. We believe this work has a positive ethical impact by contributing to more secure and reliable software development practices.

The dataset used to construct our security knowledge base is derived from publicly available sources and consists of known vulnerabilities and their fixes from CVEs and public GitHub projects. Our research does not involve human subjects or the use of personally identifiable or private data.

While any tool related to security could have potential for dual-use, our framework, RESCUE, is designed for a defensive purpose: to guide LLMs in generating secure code by leveraging knowledge of existing fixes. The methodology focuses on abstracting and applying secure coding patterns rather than discovering new exploits. We use publicly accessible LLMs and open-source tools, and our contributions aim to mitigate existing security risks in AI-assisted programming.

## REPRODUCIBILITY STATEMENT

We are committed to ensuring the reproducibility of our research. To facilitate this, we provide comprehensive resources and detailed descriptions throughout the paper. Our full implementation of the RESCUE framework is available at the repository link provided in the abstract.

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

## A    DATA COLLECTION

Our method adopts a large-scale security training dataset collected by SafeCoder (He et al., 2024). We select Python, C, and C++ instances, removing empty and duplicate instances and resulting in a raw security dataset $D$ containing 372 instances for Python and 332 instances for C/C++. Each instance includes vulnerable code, secure code, and the CWE type.

## B    METHOD DETAILS

### B.1    CLUSTER-THEN-SUMMARIZE

This appendix provides the details of the *cluster-then-summarize* pipeline for constructing compact security knowledge snippets from large collections of raw instances. The pipeline consists of two major components: (1) grouping raw instances into clusters and (2) recursively summarizing them in a bottom-up manner until a single consolidated snippet is obtained for each cluster.

**1. Cluster formation.**    Given a dataset $D$ of raw instances, we first partition $D$ into clusters $\mathcal{C} = \{C_1, C_2, \ldots, C_m\}$ based on a predefined taxonomy or grouping criterion. Each cluster $C_i$ gathers instances that share similar patterns, making it possible to produce more coherent summaries.

**2. Subset partitioning.**    Each cluster $C_i$ is further divided into fixed-size subsets of at most $b$ elements, where $b$ is a tunable parameter (default: 10). This step ensures that each subset can be fully processed within the input context of the summarizer model.

**3. First-level summarization.** For every subset $B$ within a cluster, the summarizer model $M$ is applied to generate a first-level snippet that condenses the main patterns and knowledge contained in the subset. Collecting these results yields the first-level snippet set $S_i^1 = \{s_1^1, s_2^1, \dots\}$ for cluster $C_i$.

**4. Recursive hierarchical summarization.** At each subsequent level $j \geq 2$, the set of snippets from the previous level $S_i^{j-1}$ is again partitioned into batches of size up to $b$. The summarizer model is then applied to each batch to generate a higher-level snippet. Formally,

$$S_i^j \leftarrow \bigcup_{\text{batch } B \subset S_i^{j-1}} M(B).$$

This process repeats until the number of snippets reduces to one (or a very small set), which becomes the final consolidated snippet for cluster $C_i$.

**5. Output.** The final output of the pipeline is a set of security knowledge snippets $\{k_1, k_2, \dots, k_m\}$, one for each cluster. These snippets serve as compact and generalizable abstractions distilled from raw instances.

---

**Algorithm 1:** Cluster-then-Summarize Pipeline

---

**Input:** Raw dataset $D$ of instances; batch size $b$; summarizer model $M$
**Output:** Security knowledge snippets $K = \{k_1, k_2, \dots, k_m\}$

```
// 1.  Group raw instances into clusters
```
$\mathcal{C} \leftarrow \text{Cluster}(D)$;
**foreach** *cluster $C_i \in \mathcal{C}$* **do**
    ```// 2.  Partition cluster into subsets of size up to b```
    $\mathcal{B} \leftarrow \text{Partition}(C_i, b)$;
    $S \leftarrow []$;
    **foreach** *subset $B \in \mathcal{B}$* **do**
        $s \leftarrow M.\text{Summarize}(B)$;
        $S.\text{append}(s)$;
    ```// 3.  Recursively summarize until one snippet remains```
    **while** $|S| > 1$ **do**
        $S' \leftarrow []$;
        **foreach** *batch $B_S \in Partition(S, b)$* **do**
            $s' \leftarrow M.\text{Summarize}(B_S)$;
            $S'.\text{append}(s')$;
        $S \leftarrow S'$;
    ```// 4.  Store the final snippet for cluster C_i```
    $k_i \leftarrow S[0]$;
    $K.\text{append}(k_i)$;
**return** $K$

---

The prompt for security guideline and vulnerability cause can be found at Appendix D.1 and Appendix D.2.2.

## B.2 SECURITY-FOCUSED STATIC PROGRAM SLICING

RESCUE begins by constructing a **Program Dependence Graph (PDG)** from a given code example, defined as

$$PDG = (N, E),$$

where $N$ is the set of program statements and $E$ is the set of edges comprising *data dependencies* $E_{dd}$ and *control dependencies* $E_{cd}$. Specifically, $E_{dd}$ captures relationships where a statement consumes data produced by another, while $E_{cd}$ models control-flow relationships, indicating that the execution of a statement depends on an earlier control statement, such as a conditional branch.

---

**Algorithm 2:** Bidirectional Security-Relevant Slicing in RESCUE

---

**Input:** Program Dependence Graph $PDG = (N, E)$, points of interest $\mathcal{P}_v, \mathcal{P}_s \subset N$, hop limit $h$
**Output:** Vulnerable slice $S_v$, Secure slice $S_s$
**Function** `Slice`$(PDG, \mathcal{P}, h)$**:**
    Initialize slice $S \leftarrow \emptyset$;
    **foreach** *node* $n \in N$ **do**
        **if** $\exists \pi(n, p), 1 \leq |\pi| \leq h, p \in \mathcal{P}$ **then**
            $S \leftarrow S \cup \{n\}$;
    **return** $S$;
**Function** `BidirectionalSlice`$(PDG, \mathcal{P}, h)$**:**
    $S_{back} \leftarrow$ `Slice`$(PDG, \mathcal{P}, h)$ ;               // Backward slice
    $PDG_{rev} \leftarrow$ reverse all edges in $PDG$;
    $S_{forward} \leftarrow$ `Slice`$(PDG_{rev}, \mathcal{P}, h)$ ;  // Forward slice via reversed PDG
    **return** $S_{back} \cup S_{forward}$;
$S_v \leftarrow$ `BidirectionalSlice`$(PDG, \mathcal{P}_v, h)$;
$S_s \leftarrow$ `BidirectionalSlice`$(PDG, \mathcal{P}_s, h)$;
**return** $S_v, S_s$

---

Next, RESCUE identifies **points of interest** in the code relevant to security. Deleted statements in a security patch are treated as points of interest for vulnerable code, whereas added statements indicate points of interest for secure code.

Formally, the program slicing process is modeled as a **reachability problem** over the PDG. Given a set of points of interest $\mathcal{P} \subset N$, the backward slice is computed as the set of nodes that can reach any node in $\mathcal{P}$ within $h$ hops:

$$S(PDG, \mathcal{P}, h) = \{m \in N \mid \exists \pi(m, n), 1 \leq |\pi| \leq h \ \& \ n \in \mathcal{P}\}, \tag{3}$$

where $\pi(m, n)$ is a path in the PDG, and the path length $|\pi|$ is bounded by $h$ hops.

Finally, RESCUE performs **bidirectional slicing**: backward slicing identifies nodes influencing the points of interest, and forward slicing captures statements affected by them. To ensure contextual completeness, subgraphs from vulnerable and secure code versions are compared, and each subgraph is complemented with statements from the other version outside the patch. This process reconstructs two contextually sliced code variants for secure code analysis and generation.

# C ADDITIONAL EXPERIMENTS

## C.1 ANALYSIS OF AVERAGE METHOD IMPROVEMENTS

Table 5 shows the average improvement of different methods relateive to their respective LLM Alone (Zero-Shot) baseline methods.

Table 5: Average improvement ($\Delta$) of different methods relative to their respective LLM Alone baselines. The improvements are averaged across applicable LLMs for four benchmarks: CodeGuard+, HumanEval+ (HE+), BigCodeBench (BCB), and LiveCodeBench (LCB). All values represent the mean change in percentage points.

| Method | CodeGuard+ ($\Delta$) | | | HE+ ($\Delta$) | BCB ($\Delta$) | LCB ($\Delta$) |
|---|---|---|---|---|---|---|
| | SP@1 | SR | Pass@1 | Pass@1 | Pass@1 | Pass@1 |
| SecCoder | -1.20 | 1.02 | -3.90 | -1.67 | -0.42 | 0.18 |
| Codexity | -1.67 | 6.08 | -7.43 | -0.85 | -1.10 | 0.00 |
| CoSec | -5.20 | -3.10 | -9.40 | -4.73 | -19.57 | -3.10 |
| INDICT | -23.03 | 15.88 | -43.53 | -18.40 | -20.70 | -2.45 |
| SafeCoder | 1.53 | 7.30 | -7.90 | -8.67 | -10.50 | -6.63 |
| RESCUE | 6.28 | 9.40 | -2.70 | 1.43 | -3.63 | 0.83 |

## C.2 IMPACT OF PROGRAM SLICING ON CODE LENGTH

Table 6: Average Lines of Code (LoC) for Vulnerable and Secure Samples Before and After Program Slicing. On average, program slicing reduced code lines by 81.5%, removing lines unrelated to the security aspects under consideration.

| Category | Before Slicing (Avg. LoC) | After Slicing (Avg. LoC) |
|---|---|---|
| Vulnerable Code Samples | 89.5 | 16.1 |
| Secure Code Samples | 91.3 | 17.3 |

We first performed a statistical analysis of the average number of lines of code in the raw security dataset. The findings indicate that the average length of the raw vulnerable and secure code samples was 89.5 and 91.3 lines, respectively. In contrast, after program slicing, the average lengths of the corresponding code samples were reduced to 16.1 and 17.3 lines. This reduction suggests that, on average, 81.5% of the raw code lines are unrelated to the security aspects under consideration.

## C.3 OVERHEAD AND TIME ANALYSIS

To evaluate the computational overhead of RESCUE, we conducted experiments on the CodeGuard+ benchmark under the same settings as described in the main paper. Specifically, we tested three models: Qwen2.5-Coder-7B-Instruct, Llama-3.1-8B-Instruct, and DeepSeek-V3. The first two were deployed locally using the same setup as in the paper, while DeepSeek-V3 was accessed via API. For a fair comparison, we disabled all multiprocessing operations and executed the pipeline sequentially. We then performed a fine-grained breakdown of the execution time at each step of the RESCUE online generation process.

Table 7: Average execution time (in seconds) for each step in the RESCUE online generation process.

| Model | Draft Code Generation | Vulnerability Cause Analysis | CWE-Level Retrieval | Code-Level Retrieval | Augmented Generation |
|---|---|---|---|---|---|
| Qwen2.5-Coder-7B | 2.8412 | 2.7365 | 0.0191 | 2.5892 | 3.2097 |
| Llama-3.1-8B | 3.5318 | 1.4546 | 0.0181 | 2.4340 | 3.5409 |
| DeepSeek-V3 | 12.7992 | 4.3321 | 0.0197 | 2.6314 | 10.9561 |

The results, shown in Table 7, indicate that most of the additional overhead arises from the *Draft Generation* and *Vulnerability Cause Analysis* steps, which involve multiple LLM calls. In contrast,

hierarchical retrieval is relatively lightweight: both CWE-level and code-level retrieval contribute only marginal time costs.

Overall, the overhead introduced by RESCUE is acceptable and can be further reduced. For instance, concurrent LLM calls can significantly mitigate the cost of generation and analysis steps, while efficient engineering optimizations may further improve system performance. These findings suggest that our method is scalable, and that retrieval overhead will remain manageable even on larger datasets.

### C.4  ADDITIONAL EVALUATION ON CWEVAL BENCHMARK

To further validate RESCUE beyond static analysis-based benchmarks, we evaluate it on CWEval (Peng et al., 2025), a benchmark that overcomes the limitations of static analysis tools via dynamic testing. CWEval consists of 119 high-quality security-critical coding tasks covering 31 CWEs across 5 popular programming languages. Unlike previous benchmarks that rely on static analysis tools and suffer from imprecise task specifications, CWEval employs manually crafted test oracles to capture dynamic properties of LLM-generated code, verifying both functional correctness and security against adversarial inputs. providing a more rigorous and reliable evaluation signal

We compare RESCUE against zero-shot and SecCoder across three LLMs: Qwen2.5-Coder-7B, GPT-4o-mini, and DeepSeek-V3. Since our knowledge base is constructed based on vulnerabilities in Python and C/C++, we select the subset of Python, C/C++ from CWEval. As shown in Table 8, RESCUE consistently outperforms both zero-shot and SecCoder across three models, demonstrating that our gains are not artifacts of static analysis tools or overfitting to a particular benchmark. These results further confirm the effectiveness of RESCUE.

Table 8: Comparison of RESCUE, SecCoder, and zero-shot on the CWEval benchmark, where **bold** indicates the best score.

| Model | Method | SecurePass@1 |
|---|---|---|
| Qwen2.5-Coder-7B | Zero-Shot | 36.80 |
| | SecCoder | 36.00 |
| | RESCUE | **43.70** |
| GPT-4o-mini | Zero-Shot | 51.50 |
| | SecCoder | 52.44 |
| | RESCUE | **56.40** |
| DeepSeek-V3 | Zero-Shot | 52.81 |
| | SecCoder | 61.91 |
| | RESCUE | **65.37** |

### C.5  GENERALIZABILITY ANALYSIS

#### C.5.1  CWE COVERAGE AND UNSEEN CWE TYPES

To further demonstrate the generalizability of RESCUE, we analyze the distribution of CWE types across the knowledge base and the evaluation benchmark CodeGuard+ which includes Python and C/C++. As shown in Table 9, the benchmark contains a broader variety of CWE types than the knowledge base, including many categories not present during knowledge construction. For Python and C/C++, the benchmark contains 15 and 7 unique CWE types, respectively, that are absent from the knowledge base.

Table 9: Distribution of CWE types in training set and benchmark. The benchmark contains a broader coverage of CWE types, highlighting the generalization capability of our method.

| Programming Language | # CWE Types in Training Set | # CWE Types in Benchmark | # Unique CWE Types in Benchmark |
|---|---|---|---|
| Python | 9 | 23 | 15 |
| C/C++ | 12 | 17 | 7 |

### C.5.2 BREAKDOWN AND ANALYSIS OF PERFORMANCE ON SEEN AND UNSEEN CWE TYPES

To further quantify the generalizability of RESCUE beyond the CWE types covered in its knowledge base, we break down the performance on CodeGuard+ into seen CWE types (i.e., those present in the knowledge base) and unseen CWE types (i.e., those absent from the knowledge base). The results are reported in Table 10.

As expected, RESCUE achieves larger performance gains on seen CWE types, where the retrieved knowledge directly matches the vulnerability patterns in the task. Notably, however, RESCUE still achieves a consistent average absolute improvement of 3.9 on unseen CWE types across all models, demonstrating that its benefits are not confined to memorized vulnerability patterns. We attribute this generalizability to three complementary factors. First, different CWE types often share similar vulnerability patterns — for example, various buffer overflow variants exhibit structurally analogous weaknesses that allow knowledge transfer across types. Second, the security guidelines constructed by RESCUE are inherently general and transferable, as many secure coding principles apply broadly across different CWE scenarios regardless of the specific vulnerability type. Third, since LLMs are pre-trained on large-scale code corpora, the security context retrieved by RESCUE can activate relevant parametric knowledge already encoded in the model, even when the specific CWE type was not explicitly covered during knowledge construction.

Table 10: Performance breakdown of RESCUE on seen and unseen CWE types on CodeGuard+, where $\Delta$ denotes the change relative to zero-shot.

| Model | Type | # Samples | Zero-Shot | RESCUE | $\Delta$ |
|---|---|---|---|---|---|
| DeepSeek-Coder-V2-Lite | Seen | 52 | 68.2 | 78.1 | (+9.9) |
| | Unseen | 42 | 49.3 | 50.2 | (+0.9) |
| Qwen2.5-Coder-7B | Seen | 52 | 56.6 | 76.8 | (+20.2) |
| | Unseen | 42 | 44.5 | 49.9 | (+5.4) |
| Qwen2.5-Coder-32B | Seen | 52 | 64.6 | 73.3 | (+8.7) |
| | Unseen | 42 | 52.8 | 55.0 | (+2.2) |
| Llama3.1-8B | Seen | 52 | 65.2 | 64.2 | (-1.0) |
| | Unseen | 42 | 39.5 | 46.2 | (+6.7) |
| DeepSeek-V3 | Seen | 52 | 71.1 | 76.9 | (+5.8) |
| | Unseen | 42 | 56.4 | 60.7 | (+4.3) |

### C.5.3 CROSS-LANGUAGE GENERALIZABILITY ANALYSIS

To further evaluate the generalizability of RESCUE beyond the languages covered in our knowledge base, we evaluate it on programming languages not seen during knowledge construction. Since our knowledge base is built upon vulnerabilities in Python and C/C++, we use coding tasks in Go and JavaScript from CWEval (Peng et al., 2025) to examine RESCUE's cross-language applicability. As shown in Table 11, RESCUE consistently improves over zero-shot generation across both languages and all three models, even without any language-specific security knowledge. This demonstrates that the security knowledge constructed by RESCUE is broadly transferable across programming languages, further supporting the generalizability of our approach.

Table 11: Cross-language evaluation of RESCUE on Go and JavaScript tasks from CWEval, where values in parentheses denote the gain over zero-shot.

| Model | Method | Go | JavaScript |
|---|---|---|---|
| Qwen2.5-Coder-7B | Zero-Shot | 22.81 | 50.72 |
| | RESCUE | 28.07 (+5.26) | 56.52 (+5.80) |
| GPT-4o-mini | Zero-Shot | 33.33 | 57.97 |
| | RESCUE | 38.60 (+5.27) | 62.32 (+4.35) |
| DeepSeek-V3 | Zero-Shot | 35.09 | 44.93 |
| | RESCUE | 56.14 (+21.05) | 60.87 (+15.94) |

C.6    ANALYSIS OF MODIFIED RECIPROCAL RANK FUSION (RRF)

To extend standard reciprocal rank fusion (RRF), our modified RRF introduces thresholds to filter out low-relevance retrieved results, which tend to receive low scores and may be noise for generation. To validate its effectiveness, we compare it against standard RRF on CodeGuard+ benchmark using Llama3.1-8B-Instruct and Qwen2.5-Coder-7B-Instruct, with results reported in Table 12. The results show that our modified RRF achieves better performance.

Table 12: Ablation study on fusion strategies evaluated on the CodeGuard+ benchmark, where **bold** denotes the best result.

| Model | Fusion Method | SecurePass@1 |
|---|---|---|
| Llama3.1-8B | Modified RRF | **56.2** |
| | Standard RRF | 54.8 |
| Qwen2.5-Coder-7B | Modified RRF | **64.8** |
| | Standard RRF | 57.2 |

Furthermore, to justify the choice of thresholds and examine the impact of threshold variation, we conduct a sensitivity analysis on threshold hyper-parameters. Due to computational constraints, we randomly sample 20 instances from CodeGuard+ as validation data and evaluate Llama3.1-8B-Instruct and Qwen2.5-Coder-7B-Instruct. By fixing two thresholds and varying the third, Table 13, 14, and 15 show the results, where numbers in parentheses denote the change relative to zero-shot. The overall trend remains stable, confirming the robustness of threshold selection.

Table 13: Sensitivity analysis of the API threshold, where values in parentheses denote the change relative to the zero-shot baseline.

| Model | API Thr. | SecurePass@1 | Pass@1 | SecureRate |
|---|---|---|---|---|
| Llama3.1-8B | 4 (default) | 56.7 (+8.4) | 86.7 (-1.6) | 66.7 (+11.7) |
| | 0.0 | 51.7 (+3.4) | 81.7 (-6.6) | 65.0 (+10.0) |
| | 2.0 | 53.3 (+5.0) | 86.7 (-1.6) | 65.0 (+10.0) |
| | 3.0 | 53.3 (+5.0) | 78.3 (-10.0) | 66.7 (+11.7) |
| | 5.0 | 51.7 (+3.4) | 83.3 (-5.0) | 65.0 (+10.0) |
| | 7.0 | 50.0 (+1.7) | 80.0 (-8.3) | 65.0 (+10.0) |
| Qwen2.5-Coder-7B | 4 (default) | 61.7 (+5.0) | 91.7 (-1.6) | 65.0 (+6.7) |
| | 0 | 59.5 (+2.8) | 94.5 (+1.2) | 63.7 (+5.4) |
| | 2 | 61.0 (+4.3) | 94.5 (+1.2) | 62.9 (+4.6) |
| | 3 | 58.5 (+1.8) | 92.0 (-1.3) | 61.8 (+3.5) |
| | 5 | 62.0 (+5.3) | 94.5 (+1.2) | 64.5 (+6.2) |
| | 7 | 62.5 (+5.8) | 94.5 (+1.2) | 64.2 (+5.9) |

Table 14: Sensitivity analysis of the Vulnerability Cause Analysis (VCA) threshold, where values in parentheses denote the change relative to the zero-shot baseline.

| Model | VCA Thr. | SecurePass@1 | Pass@1 | SecureRate |
|---|---|---|---|---|
| Llama3.1-8B | 0.75 (default) | 56.7 (+8.4) | 86.7 (-1.6) | 66.7 (+11.7) |
| | 0.00 | 46.7 (-1.6) | 80.0 (-8.3) | 61.7 (+6.7) |
| | 0.25 | 50.0 (+1.7) | 81.7 (-6.6) | 63.3 (+8.3) |
| | 0.45 | 46.7 (-1.6) | 76.7 (-11.6) | 63.3 (+8.3) |
| | 0.65 | 55.0 (+6.7) | 88.3 (0.0) | 61.7 (+6.7) |
| | 0.85 | 45.0 (-3.3) | 75.0 (-13.3) | 58.3 (+3.3) |
| | 0.95 | 51.7 (+3.4) | 86.7 (-1.6) | 60.0 (+5.0) |
| Qwen2.5-Coder-7B | 0.75 (default) | 61.7 (+5.0) | 91.7 (-1.6) | 65.0 (+6.7) |
| | 0.00 | 58.5 (+1.8) | 90.5 (-2.8) | 64.5 (+6.2) |
| | 0.25 | 62.0 (+5.3) | 92.5 (-0.8) | 64.4 (+6.1) |
| | 0.45 | 60.0 (+3.3) | 93.5 (+0.2) | 61.0 (+2.7) |
| | 0.65 | 61.5 (+4.8) | 95.0 (+1.7) | 64.0 (+5.7) |
| | 0.85 | 60.0 (+3.3) | 95.5 (+2.2) | 62.4 (+4.1) |
| | 0.95 | 58.0 (+1.3) | 93.5 (+0.2) | 61.8 (+3.5) |

Table 15: Sensitivity analysis of the Code threshold, where values in parentheses denote the change relative to the zero-shot baseline.

| Model | Code Thr. | SecurePass@1 | Pass@1 | SecureRate |
|-------|-----------|--------------|--------|------------|
| Llama3.1-8B | 0.65 (default) | 56.7 (+8.4) | 86.7 (-1.6) | 66.7 (+11.7) |
| | 0.00 | 55.0 (+6.7) | 80.0 (-8.3) | 65.0 (+10.0) |
| | 0.15 | 55.0 (+6.7) | 83.3 (-5.0) | 65.0 (+10.0) |
| | 0.35 | 50.0 (+1.7) | 81.7 (-6.6) | 66.7 (+11.7) |
| | 0.55 | 51.7 (+3.4) | 80.0 (-8.3) | 65.0 (+10.0) |
| | 0.75 | 46.7 (-1.6) | 70.0 (-18.3) | 66.7 (+11.7) |
| | 0.85 | 45.0 (-3.3) | 75.0 (-13.3) | 63.3 (+8.3) |
| | 0.95 | 46.7 (-1.6) | 75.0 (-13.3) | 65.0 (+10.0) |
| Qwen2.5-Coder-7B | 0.65 (default) | 61.7 (+5.0) | 91.7 (-1.6) | 65.0 (+6.7) |
| | 0.00 | 59.5 (+2.8) | 92.0 (-1.3) | 60.4 (+2.1) |
| | 0.15 | 62.0 (+5.3) | 95.5 (+2.2) | 62.7 (+4.4) |
| | 0.35 | 61.0 (+4.3) | 97.5 (+4.2) | 61.6 (+3.3) |
| | 0.55 | 61.0 (+4.3) | 93.5 (+0.2) | 61.7 (+3.4) |
| | 0.75 | 60.0 (+3.3) | 92.5 (-0.8) | 60.0 (+1.7) |
| | 0.85 | 60.0 (+3.3) | 93.0 (-0.3) | 61.1 (+2.8) |
| | 0.95 | 61.5 (+4.8) | 94.5 (+1.2) | 61.8 (+3.5) |

## C.7 ANALYSIS OF FUNCTIONALITY-SPECIFIC AND NON-FUNCTIONALITY-SPECIFIC VULNERABILITIES

To further understand the performance of RESCUE across different vulnerability types, we follow Chen et al. (2025) to classify the tasks in CodeGuard+ into functionality-specific and non-functionality-specific categories. Specifically, functionality-specific vulnerabilities refer to flaws that arise from unsafe or incorrect use of APIs and library-dependent logic, such as SQL injection, XXE, command injection, and deserialization. Non-functionality-specific vulnerabilities, by contrast, refer to memory-related flaws that violate low-level programming constraints, such as buffer overflows and null pointer dereferences. We analyze the performance breakdown of RESCUE under each category, with results reported in Table 16.

As shown in Table 16, various models generally achieve higher scores on non-functionality-specific vulnerabilities than on functionality-specific ones. Compared to the zero-shot baseline, RESCUE yields a larger average performance gain on non-functionality-specific vulnerabilities (+7.88) than on functionality-specific vulnerabilities (+6.02).

We attribute this performance difference to the inherent nature of the two vulnerability categories. Non-functionality-specific vulnerabilities typically involve checking explicit and local constraints, such as buffer sizes and null pointer dereferences. Such knowledge can be more readily extracted during the knowledge retrieval phase and effectively incorporated into the code generation process. In contrast, functionality-specific vulnerabilities arise from unsafe API usage and implicit threat models tied to specific libraries, rather than violations of generic programming logic. Addressing these vulnerabilities requires concrete, domain-specific knowledge that is inherently harder to retrieve and apply. This explains why zero-shot models struggle more with functionality-specific tasks, and why RESCUE achieves a comparatively smaller performance gain in this category.

Table 16: Performance breakdown by vulnerability type (functionality-specific vs. non-functionality-specific) on CodeGuard+, where $\Delta$ denotes the change relative to the zero-shot baseline.

| Model | Type | # Samples | Zero-Shot | RESCUE | $\Delta$ |
|-------|------|-----------|-----------|--------|----------|
| DeepSeek-Coder-V2-Lite | Func | 67 | 54.8 | 56.7 | (+1.9) |
|  | Non-Func | 27 | 72.0 | 87.7 | (+15.7) |
| Qwen2.5-Coder-7B | Func | 67 | 41.1 | 54.5 | (+13.4) |
|  | Non-Func | 27 | 76.2 | 90.3 | (+14.1) |
| Qwen2.5-Coder-32B | Func | 67 | 51.0 | 55.5 | (+4.5) |
|  | Non-Func | 27 | 80.1 | 88.9 | (+8.8) |
| Llama3.1-8B | Func | 67 | 48.4 | 49.3 | (+0.9) |
|  | Non-Func | 27 | 66.9 | 73.3 | (+6.4) |
| DeepSeek-V3 | Func | 67 | 53.6 | 63.0 | (+9.4) |
|  | Non-Func | 27 | 91.9 | 86.3 | (-5.6) |

# D PROMPT TEMPLATE

This appendix section describes the details of our used prompt templates.

## D.1 SECURITY GUIDELINES EXTRACTION

### D.1.1 INITIAL GUIDELINE SUMMARIZATION

Listing 1: This prompt describes how to extract initial security guidelines at the bottom.

```
Extract common security knowledge from the provided multiple cases
    . Identify and summarize distinctive guidelines.

# Output Format
Provide a clear and concise summary of each guideline in a
    sentence. Ensure that each guideline is distinct.

# Output Example
- Use parameterized queries instead of string interpolation or
    concatenation to prevent SQL injection attacks.

Extract security guidelines from the following cases:

# Vulnerability-Fix Data List

## Case 1
### Vulnerable Code
{vulnerable code 1}

### Security Patch
{security patch 1}

## Case 2
### Vulnerable Code
{vulnerable code 2}

### Security Patch
{security patch 2}

...
```

### D.1.2 INTERMEDIATE GUIDELINE SUMMARIZATION

Listing 2: This prompt is used to iteratively summarize the intermediate security guidelines.

```
Merge extracted security guidelines by combining those with
    identical meanings into a single, clear, and concise sentence
    for each guideline.

# Extracted Guidelines 1
{intermediate guidelines 1}

# Extracted Guidelines 2
{intermediate guidelines 2}
...
```

## D.2    PROACTIVE ANALYSIS

### D.2.1    ZERO-SHOT CODE GENERATION

The completion models directly use the target coding task as prompt.

Listing 3: The chat models—Deepseek-V3 and GPT-4o-mini—employ this prompt template for zero-shot code generation.

```
Generate the following {lang} code.

### Code Snippet and Task Requirement
```{lang}
{code}
```
```

### D.2.2    VULNERABILITY CAUSE ANALYSIS

Listing 4: This prompt is used to analyze and identify the potential vulnerability cause in advance.

```
Identify and summarize the cause of the potential vulnerability in
    a concise and informative paragraph before proceeding with
    code generation. The output should at least include the name
    of the vulnerability and its cause. Focus solely on the
    underlying cause of the vulnerability without mentioning any
    fixing information.
The output should be a text-only paragraph, without any other
    structure.

# Example
** Input **:
```python
def build_board(conn, game,size):
    # we'll build the empty board, and then fill in with the move
        list that
    # we get from the DB.
    board = []
    for i in range(size):
        board.append([""]*size)

    # search for all moves that have happenend during this game.
    cursor = conn.cursor()
```

** Output **:
```

```
The potential vulnerability is related to an SQL Injection
    vulnerability. This vulnerability occurs because the code
    implies the execution of SQL queries based on user-supplied
    inputs without implementing proper validation or sanitization
    mechanisms. The use of `conn.cursor()` to execute presumably
    dynamic SQL commands indicates that if user inputs are
    directly incorporated into those commands, it could lead to
    unauthorized access or manipulation of the database. This
    vulnerability is typically exploited by injecting malicious
    SQL statements through user inputs, which the application
    fails to properly handle or escape, thereby compromising the
    database's integrity and security.

Please identify and summarize the task below:
```{lang}
{code}
```
```

## D.3  SECURITY KNOWLEDGE AUGMENTED CODE GENERATION

Listing 5: This prompt is used for completion models, including all locally deployed models.

```
# General Security Guidelines
```
{security_guidelines}
```

# Example
## Secure Code Example
```{language}
{secure_code}
```

## Secure Code Example
```{lang}
{code}
```

Listing 6: This prompt is used for chat models, including GPT-4o-mini and Deepseek-V3.

```
Given the security knowledge:
```markdown
# General Security Guidelines
```
{security_guidelines}
```

# Example
## Secure Code Example
```{language}
{secure_code}
```
```

Your task is to complete the following {lang} code.
### Code Snippet
```{lang}
{code}
```
```

```
When completing, you should consider the following:
1. You must not change the code snippet part in the completed code
   , including the function signature, import statements.
2. You can refer to the provided security knowledge but not simply
    copy and paste. You should first think how they work and if
   they can be applied to the code snippet.

Your response should start with ```{lang}
```

## E  VULNERABILITY-FIX INSTANCE EXAMPLE

Listing 7: The complete example of a vulnerability-fix instance. the red highlighted text means the deleted statements and the gree highlighted text means the added statements. This instance has about 50 lines of code. However, it contains only one line related to security. And the majority of the function is to implement the database query.

```python
--- vulnerability.py

+++ fixing.py

@@ -1,55 +1,55 @@

 @app.route('/puppet/default', methods=['GET', 'POST'])
 @cortex.lib.user.login_required
 def puppet_enc_default():
   """Handles the Puppet ENC Default Classes page"""

   # Check user permissions
   if not does_user_have_permission("puppet.default_classes.view"):
     abort(403)

   # Get the default YAML out of the kv table
   curd = g.db.cursor(mysql.cursors.DictCursor)
   curd.execute("SELECT `value` FROM `kv_settings` WHERE `key` = 'puppet.enc.default'")
   result = curd.fetchone()
   if result == None:
     classes = "# Classes to include on all nodes using the default settings can be entered
         here\n"
   else:
     classes = result['value']

   # On any GET request, just display the information
   if request.method == 'GET':
     return render_template('puppet/default.html', classes=classes, active='puppet', title="
         Default Classes")

   # On any POST request, validate the input and then save
   elif request.method == 'POST':
     # Check user permissions
     if not does_user_have_permission("puppet.default_classes.edit"):
       abort(403)

     # Extract data from form
     classes = request.form.get('classes', '')

     # Validate classes YAML
     try:
-      data = yaml.load(classes)
+      data = yaml.safe_load(classes)
     except Exception as e:
       flash('Invalid YAML syntax: ' + str(e), 'alert-danger')
       return render_template('puppet/default.html', classes=classes, active='puppet', title="
           Default Classes")

     try:
       if not data is None:
         assert isinstance(data, dict)
     except Exception as e:
       flash('Invalid YAML syntax: result was not a list of classes, did you forget a trailing
             colon? ' + str(e), 'alert-danger')
       return render_template('puppet/default.html', classes=classes, active='puppet', title="
           Default Classes")
```

```
# Get a cursor to the database
# Update the system
curd.execute('REPLACE INTO `kv_settings` (`key`, `value`) VALUES ("puppet.enc.default", %
    s)', (classes,))
g.db.commit()

cortex.lib.core.log(__name__, "puppet.defaultconfig.changed", "Puppet default
    configuration updated")
# Redirect back
flash('Puppet default settings updated', 'alert-success')

return redirect(url_for('puppet_enc_default'))
```

## F  LIMITATIONS

**Static Analysis-Based Evaluation.**  Our evaluation primarily relies on static security analysis tools, which are prone to false positives and negatives. In particular, complex inter-procedural analyses may not be fully captured, potentially leading to discrepancies in evaluation results. While we additionally evaluate RESCUE on CWEval in Appendix C.4, which employs dynamic testing, broader adoption of dynamic evaluation methods remains an important direction for future work.

**Function-Level Coding Tasks.**  RESCUE currently focuses on function-level code generation. Extending it to large-scale, repository-level deployment scenarios requires further adaptation, such as handling cross-file dependencies and longer code contexts. We consider this a promising direction for future work.

**Inference Overhead.**  RESCUE introduces additional inference overhead compared to zero-shot generation. This overhead mainly stems from the proactive analysis steps, which require additional LLM calls. We note that many modern coding agents already incur comparable and often greater overhead due to complex reasoning steps, and the security improvements achieved by RESCUE — an average of 6.3% absolute gain in SecurePass@1 — justify this cost, given that security vulnerabilities in production can result in substantial consequences. Nevertheless, several directions exist to mitigate this overhead, including parallelizing independent LLM calls and introducing a lightweight security-relevance filter to selectively invoke RESCUE only on tasks that involve potentially vulnerable patterns.

## G  THE USE OF LARGE LANGUAGE MODELS (LLMS)

In preparing this paper, we used a large language model (LLM) solely as a writing assistant for text refinement. Specifically, the LLM was used to polish grammar, improve clarity, and adjust wording for better readability.

