# OpenReview forum: "RESCUE: Retrieval Augmented Secure Code Generation"
_ICLR.cc/2026/Conference — ICLR 2026 Poster_

### Official Review · Reviewer_M2w4 · 2025-10-28

**Soundness:** 3
**Presentation:** 3
**Contribution:** 2
**Rating:** 2
**Confidence:** 4

**Summary:**

This paper introduces RESCUE, a retrieval-augmented framework designed to enhance the security of code generated by LLMs. The framework constructs a hybrid knowledge base by combining LLM-summarized security guidelines with statically extracted code slices and employs a hierarchical retrieval process across three facets. Experiments across six LLMs and four benchmarks demonstrate consistent improvements in both functionality and security, including a 4.8% gain in SecurePass@1 on CodeGuard+. The work effectively integrates static analysis with retrieval to guide more secure code generation.

**Strengths:**

- Addresses a timely and important problem concerning security vulnerabilities in code generated by large language models.
- Demonstrates strong empirical results through extensive experiments across multiple models and benchmarks.
- The paper is clearly written and well-organized.

**Weaknesses:**

- Incremental contribution relative to prior work.
While RESCUE is a well-engineered and empirically validated framework, its core ideas appear incremental relative to existing work in secure code generation. Several prior studies have already applied retrieval-augmented generation (RAG) to integrate security knowledge into large language models. For instance, CodeGuarder [1] employs security-enhanced retrieval to inject vulnerability knowledge and code examples, while Tony et al. [2] dynamically retrieve CWE-based secure coding guidelines to mitigate unsafe generations. Furthermore, hierarchical RAG structures similar to RESCUE’s have been explored in HiRAG [3]. As a result, the primary novelty of RESCUE seems to lie in method integration and empirical comprehensiveness, rather than methodological innovation. The authors should more clearly articulate the conceptual distinctions between RESCUE and these prior approaches and include them as baselines to substantiate the claimed advantages.

>[1] Lin, Bo, et al. “Give LLMs a Security Course: Securing Retrieval-Augmented Code Generation via Knowledge Injection.” arXiv preprint arXiv:2504.16429 (2025).

>[2] Tony, Catherine, Emanuele Iannone, and Riccardo Scandariato. “Retrieve, Refine, or Both? Using Task-Specific Guidelines for Secure Python Code Generation.” ICSME 2025.

>[3] Huang, Haoyu, et al. “Retrieval-Augmented Generation with Hierarchical Knowledge.” arXiv preprint arXiv:2503.10150 (2025).

- Limited scale and coverage of the security knowledge base.
The security knowledge base used by RESCUE is derived from SafeCoder, containing only 372 Python and 332 C/C++ instances. This limited scale raises concerns regarding coverage, representativeness, and generalization, especially given the diversity and complexity of real-world software ecosystems. The authors should discuss the cross-domain applicability of RESCUE, its robustness to unseen frameworks or programming languages, and how the restricted dataset size may constrain retrieval effectiveness.

- Unclear design of recursive summarization pipeline.
The recursive summarization pipeline groups CWE clusters into batches of ≤10 instances, but the paper does not specify whether batching is guided by semantic similarity, class balance, or representativeness. If batches are formed randomly, distinct vulnerability–fix pairs may be merged, yielding over-generalized or contradictory summaries. Fixed batch sizes may also introduce information imbalance, causing rare yet critical vulnerabilities to be underrepresented. Such an uncontrolled summarization process risks semantic drift and bias in the resulting knowledge base.

- Insufficient justification of rank fusion method.
Although the paper identifies rank fusion as a key innovation, its mathematical formulation and empirical validation are insufficiently detailed. The “modified RRF” method is described only through a formula involving thresholding and rank filtering, with hyperparameters stated to follow prior work. However, no ablation study isolates its contribution or demonstrates how it improves retrieval quality compared to standard RRF. As presented, the approach reads as a heuristic adjustment rather than a rigorously justified advancement.

**Questions:**

1. How does RESCUE fundamentally differ from prior RAG-based systems such as CodeGuarder or HiRAG?
2. Can RESCUE generalize effectively to diverse real-world vulnerabilities beyond its limited training set?
3. What is the computational cost of RESCUE in realistic code-generation scenarios, and how feasible is its deployment at scale?

---

> ### Author Response · Authors · 2025-11-25
> **Response to Reviewer M2w4 (1/n)**
>
> Dear Reviewer M2w4:
>
> Thank you for your time and effort in reviewing our work. We answer your questions below.
>
> > W1\&Q1: Difference between prior RAG-based work and RESCUE.
>
> We cited several RAG-based secure code generation methods and discussed how RESCUE differs from them (Section 1 Lines 49-70  and Section 4 Lines 472-476). But thank you for mentioning the additional papers \[1, 2, 3\]. We were not aware of them before since they were more recent work in parallel to RESCUE and were posted on arXiv in 2025\.
>
> After checking these papers, we noticed several key differences between RESCUE and them. We summarize these differences below and will discuss them in the revised paper.
>
> **(a) Comparison to CodeGuarder \[1\]**
>
> 1. CodeGuarder constructs a security knowledge base by extracting security knowledge from security-related code examples. There is no hierarchical structure in the knowledge base. Furthermore, RESCUE addresses a unique challenge in RAG-based secure code generation—program logic from a retrieved code example that is irrelevant to the target programming task may distract the LLM and produce secure but functionally incorrect code. Specifically, RESCUE performs program slicing to only retain security-related code semantics in a code example when constructing the knowledge base. The ablation study shows that without program slicing, the functional correctness of generated code drops significantly as measured by P@1 in Table 2\. We are not aware of any other work that performs such a denoising process in security knowledge curation.
> 2. CodeGuarder and RESCUE adopt very different retrieval methods. CodeGuarder decomposes a code generation query into sub-tasks and performs embedding-based retrieval based on the functional requirement of each subtask. By contrast, RESCUE performs a multi-faceted analysis first by explicitly reasoning about three security-critical aspects: vulnerability cause analysis, API patterns, and code similarity (Section 2.2.1). Based on the analysis results, RESCUE further performs a hierarchical retrieval and fuses the results retrieved from these three aspects. We are also not aware of any other work that performs such a multifaceted security analysis and retrieval.
>
>
> **(b) Comparison to Tony et al. \[2\]**
>
> 1. While Tony et al. leverage CWE-based secure coding guidelines, these guidelines were manually constructed by the authors. As reported in the paper, it took 100 hours to construct 320 guidelines. By contrast, RESCUE automatically summarizes security guidelines based on security patches made to vulnerable code instances with the same root cause (i.e., CWE).
> 2. Tony et al. use standard vector search in RAG to retrieve related security guidelines. However, RESCUE performs multi-faceted analysis and hierarchical retrieval, as detailed in the comparison to CodeGuarder.
>
> **(c) Comparison to HiRAG \[3\]**
> Although HiRAG \[3\] also uses the term “hierarchical knowledge”, it addresses a fundamentally different problem with a very different approach compared to RESCUE.
>
> 1. Regarding the research problem, HiRAG aims to address the knowledge gap between local knowledge (i.e., retrieved entities) and global knowledge (i.e., retrieved communities) in Graph RAG. However, RESCUE aims to address data noise in security knowledge bases and the limitation of conventional retrievers in security knowledge retrieval (Lines 51-71).
> 2. Regarding the technical approach, HiRAG introduces a bridge layer between the local knowledge layer and the global knowledge layer by creating "shortcuts" between entities that are distantly located in lower layers. By contrast, RESCUE performs program slicing to remove code unrelated to security and also performs multi-faceted analysis to more accurately reason about different security aspects of a coding task and identify relevant security knowledge.
>
> \[1\] Lin, Bo, et al. “Give LLMs a Security Course: Securing Retrieval-Augmented Code Generation via Knowledge Injection.” arXiv preprint arXiv:2504.16429 (2025).
>
> \[2\] Tony, Catherine, Emanuele Iannone, and Riccardo Scandariato. “Retrieve, Refine, or Both? Using Task-Specific Guidelines for Secure Python Code Generation.” ICSME 2025\.
>
> \[3\] Huang, Haoyu, et al. “Retrieval-Augmented Generation with Hierarchical Knowledge.” arXiv preprint arXiv:2503.10150 (2025).

---

> ### Author Response · Authors · 2025-11-25
> **Response to Reviewer M2w4 (2/n)**
>
> > W2 & Q2: Limited CWE coverage of the current knowledge base & Cross-domain applicability
>
> Since the offline knowledge construction process from RESCUE is fully automated. We can easily expand the knowledge base to cover more vulnerability instances and types as more data becomes available. Note that our contribution here is not the knowledge base, but rather the automated method to build the knowledge base.
>
> Furthermore, Appendix B.4 shows that the security benchmark used in our evaluation, CodeGuard+,  includes many CWE types that do not appear in the knowledge base of RESCUE. This indicates the generalizability of RESCUE beyond seen CWE types. To examine this further, we break down the results into seen and unseen CWE types.  As shown in the following table, for unseen CWE types, RESCUE can still achieve an absolute improvement of 3.9 on average. We suppose this generalizability comes from several factors:
>
> 1. Different CWE types, such as different types of buffer overflows, could share similar vulnerability patterns.
> 2. Security guidelines are common and universal even for different CWE scenarios.
> 3. LLMs are trained on massive code data, meaning that the security context can activate relevant parametric knowledge in LLMs.
>
> | Model | Type | \# Samples | Zero Shot | RESCUE | $\Delta$ |
> | :---- | :---- | :---- | :---- | :---- | :---- |
> | DeepSeek-Coder-V2-Lite | seen | 52 | 68.2 | 78.1 | \+9.9 |
> |  | unseen | 42 | 49.3 | 50.2 | \+0.9 |
> | Qwen2.5-Coder-7B | seen | 52 | 56.6 | 76.8 | \+20.2 |
> |  | unseen | 42 | 44.5 | 49.9 | \+5.4 |
> | Qwen2.5-Coder-32B | seen | 52 | 64.6 | 73.3 | \+8.7 |
> |  | unseen | 42 | 52.8 | 55.0 | \+2.2 |
> | Llama3.1-8B | seen | 52 | 65.2 | 64.2 | \-1.0 |
> |  | unseen | 42 | 39.5 | 46.2 | \+6.7 |
> | DeeSeek-V3 | seen | 52 | 71.1 | 76.9 | \+5.8 |
> |  | unseen | 42 | 56.4 | 60.7 | \+4.3 |
>
> Finally, we evaluate RESCUE on programming languages that are not covered in our current knowledge base. Since our knowledge base is constructed based on vulnerabilities in Python and C/C++, we use coding tasks in Go and JavaScript from CWEval \[4\] to evaluate RESCUE’s cross-language applicability . The results are shown in the table below. RESCUE still provides noticeable improvements even without language-specific knowledge. This suggests that the security knowledge we construct is broadly applicable across languages and further supports the generalizability of our method.
>
> | Model | Method | Go | Javascript |
> | :---- | :---- | :---- | :---- |
> | Qwen2.5-Coder-7B | Zero Shot | 22.81 | 50.72 |
> |  | RESCUE | 28.07(+5.26) | 56.52(+5.8) |
> | GPT-4o-mini | Zero Shot | 33.33 | 57.97 |
> |  | RESCUE | 38.6(+5.27) | 62.32(+4.35) |
> | DeepSeek-V3 | Zero Shot | 35.09 | 44.93 |
> |  | RESCUE | 56.14(+21.05) | 60.87(+15.94) |
>
> \[4\] Peng, Jinjun, Leyi Cui, Kele Huang, Junfeng Yang, and Baishakhi Ray. "Cweval: Outcome-driven evaluation on functionality and security of llm code generation." In 2025 IEEE/ACM International Workshop on Large Language Models for Code (LLM4Code), pp. 33-40. IEEE, 2025\.

---

> ### Author Response · Authors · 2025-11-25
> **Response to Reviewer M2w4 (n/n)**
>
> > W3: Unclear design of the recursive summarization pipeline.
>
> We would like to clarify that our summarization method operates on data samples already grouped by CWE type, meaning that all instances within a cluster share the same root cause and vulnerability pattern. This yields strong intrinsic semantic homogeneity, and therefore the batched samples within each CWE group already exhibit high security relevance. As a result, complex batching strategies (e.g., semantic similarity sorting) are unnecessary and random batching is sufficient in this setting. Furthermore, our recursive design ensures that every data instance contributes to the bottom-up summarization process, preventing the underrepresentation of rare cases. To explicitly mitigate the risk of semantic drift or contradictions, our prompt design (Appendix D.1.1) instructs the model to "Extract common security knowledge" and "Identify and summarize distinctive guidelines". This ensures that the final knowledge base represents a comprehensive synthesis of all valid fixing strategies within a CWE type, rather than a biased selection.
>
> > W4: Justification of the rank fusion method.
>
> We extend the standard Reciprocal Rank Fusion (RRF) with thresholds for filtering. Standard RRF aggregates multiple ranked lists via the rank of each item $\sum_i \frac{1}{r_i(d)+\alpha}$, so that higher-ranked items across facets receive larger contributions, providing a simple and robust way. However, in our setting not every facet should influence the fusion results. Specifically, low score results may be noise for generation. Thus, we introduce a gating term $V_i(d)=\mathbb{I}(s_i(d)>\tau_i)\cdot\mathbb{I}(r_i(d)\le 10)$ and compute the fused score as $\mathrm{RRF}(d)=\sum_{i=1}^{f} V_i(d)\cdot \frac{1}{r_i(d)+\alpha}$. This modification preserves the intuition of standard RRF while ensuring that only high-confidence, top-ranked facets contribute, effectively filtering out irrelevant or low-quality retrieval results. To prove its effectiveness, we compare it with standard RRF on CodeGuard+ benchmark below.
>
> | Model | Fusion Method | SecurePass@1 |
> | :---- | :---- | ----- |
> | Llama3.1-8B | Modified RRF | 56.2 |
> |  | Standard RRF | 54.8 |
> | Qwen2.5-Coder-7B | Modified RRF | 64.8 |
> |  | Standard RRF | 57.2 |
>
> > Q3: Practical deployment in real-world scenarios.
>
> Thank you for this question. Appendix B.3 provides a detailed analysis of the time cost of individual steps of RESCUE. Depending on the underlying LLM used in RESCUE, the total inference time of RESCUE per coding task ranges from 10 seconds to 30 seconds. Many coding agents these days already perform complex reasoning steps in code generation, which also introduces similar runtime overhead. So we think this additional time cost is reasonable and won’t lead to a significant slowdown in the developer workflow.
>
> Regarding deployment at scale, our offline pipeline is fully automated, allowing it to efficiently construct security knowledge bases from large-scale datasets without manual intervention. Furthermore, our hierarchical design ensures online scalability by effectively partitioning the search space into smaller, relevant subsets, which optimizes retrieval efficiency even as the underlying data grows. These design choices make RESCUE a practical and scalable solution for realistic deployment scenarios.

---

### Official Review · Reviewer_WRYK · 2025-10-29

**Soundness:** 2
**Presentation:** 3
**Contribution:** 2
**Rating:** 6
**Confidence:** 4

**Summary:**

This paper proposes \textsc{Rescue}, a hybrid knowledge base construction method that combines LLM-assisted cluster-then-summarize distillation with program slicing, producing both high-level security guidelines and concise, security-focused code examples. It designs a hierarchical multi-faceted retrieval to traverse the constructed knowledge base from top to bottom and integrates multiple security-critical facts at each hierarchical level.
This paper evaluates \textsc{Rescue} on four benchmarks and compared it with five state-of-the-art secure code generation methods on six LLMs.
Its evaluation results demonstrate that \textsc{Rescue} improves the SecurePass@1 metric by an average of 4.8 points.

**Strengths:**

- This paper focuses on addressing an important question
- This paper's results has shown substantially improvement compared with baselines
- This paper design a comprehensive retrieval approach for related vulnerabilities

**Weaknesses:**

Generally, the paper tries to address an important security problem while some concerns about the evaluation setting exist.

1.1 Evaluation Benchmark. The main concern of this paper is that the evaluation benchmarks are programming-contest benchmarks (HE, BCB, LCB). These benchmarks are mainly self-contained, and mostly function-level. Avoiding vulnerabilities in these benchmarks are not convincing and this paper can be substantially improved after including real-world code-generation/-completion benchmarks.

1.2 Additionally, the retrieved dataset consists only real-world (repo-level) vulnerabilities and their patches. Thus, this paper needs to add necessary motivation examples to illustrate how these retrieved examples help address potential vulnerabilities.
Please note that Figure 2 is not convincing. In a HumanEval/Livecodebench task, it is usually meaningless to replace `yaml.load()` with `yaml.safe_load()`.

1.3 Vulnerability Selection. The SafeCoder dataset consists both C/C++ and Python vulnerabilities while the evaluation is conducted on only Python (HE, BCB, LCB).

2. Note that vulnerabilities can be divided into functionality-specific (e.g., SQL injection) and non-functionality-specific (i.e., memory-related ones) [1], this paper is suggested to add a deep analysis about how various CWE types are used in retrieval and their contribution to the end-to-end improvement.

[1] Chen T, Wang Z, Li L, et al. Detecting Functionality-Specific Vulnerabilities via Retrieving Individual Functionality-Equivalent APIs in Open-Source Repositories[C]//39th European Conference on Object-Oriented Programming (ECOOP 2025). Schloss Dagstuhl–Leibniz-Zentrum für Informatik, 2025: 6: 1-6: 27.

**Questions:**

Please refer to the preceding questions.

---

> ### Author Response · Authors · 2025-11-25
> **Response to Reviewer WRYK (1/2)**
>
> Dear Reviewer WRYK:
>
> Thank you for your comments and feedback\! We hope our responses address your concerns.
>
> > W1.1: Evaluation benchmark.
>
> Thank you for raising this question. Our primary goal is to enhance the security of code generated by LLMs. However, prior work \[1\] noted that improving security often compromises functional correctness. To evaluate this tradeoff, we include HE, BCB, and LCB, which are widely used for evaluating functional correctness of LLM-generated code. Nevertheless, our primary benchmark is a security-sensitive benchmark called CodeGuard+. CodeGuard+ originates from real-world security scenarios and contains coding tasks that reflect various types of real-world vulnerabilities.
>
> Following your suggestion, we further evaluated RESCUE with another security-sensitive benchmark, CWEval \[2\]. Compared with existing benchmarks, CWEval uses dynamic testing to jointly evaluate the functionality and security. Due to the time limit, we have only compared RESCUE with zero-shot generation and SecCoder across three LLMs. The results are shown below. RESCUE consistently outperforms zero-shot generation and SecCoder across three LLMs. We believe this benchmark further strengthens our evaluation. We will incorporate these results in our next revision.
>
> | Model | Method | SecurePass@1 |
> | :---- | :---- | :---- |
> | Qwen2.5-Coder-7B | Zero Shot | 36.8 |
> |  | SecCoder | 36.0 |
> |  | RESCUE | **43.7** |
> | GPT-4o-mini | Zero Shot | 51.5 |
> |  | SecCoder | 52.44 |
> |  | RESCUE | **56.4** |
> | Deepseek-V3 | Zero Shot | 52.81 |
> |  | SecCoder | 61.91 |
> |  | RESCUE | **65.37** |
>
> \[1\] He, Jingxuan, Mark Vero, Gabriela Krasnopolska, and Martin Vechev. "Instruction Tuning for Secure Code Generation." In International Conference on Machine Learning, pp. 18043-18062. PMLR, 2024\.
>
> \[2\] Peng, Jinjun, Leyi Cui, Kele Huang, Junfeng Yang, and Baishakhi Ray. "Cweval: Outcome-driven evaluation on functionality and security of llm code generation." In 2025 IEEE/ACM International Workshop on Large Language Models for Code (LLM4Code), pp. 33-40. IEEE, 2025\.
>
> > W1.2 Motivation examples.
>
> Thank you for this suggestion. Previously, we used Figure 2 only to illustrate the need for program slicing and did not clearly demonstrate how the retrieved example helps prevent the generation of vulnerable code. We will add the following coding task to demonstrate this.
>
> ```python
> @app.route('/load')
> def load():
>     '''
>     Use the filename variable to open a file,
>     load its contents into a variable with yaml, and return it in the json format
>     '''
>     filename \= request.args.get('filename', None)
> ```
>
> This task requires loading a YAML file on the backend. A vulnerable implementation with `yaml.load()` can lead to code execution vulnerability in the production server. In such cases, our retrieved example in Figure 2 provides concrete guidance on the secure way to process untrusted data, helping the LLMs to avoid generating insecure code.
>
> > W1.3 Programming language selection.
>
> Thank you for this question. We would like to clarify that the CodeGuard+ benchmark already includes both Python and C/C++ samples for evaluation. The newly added benchmark, CWEval, is also multilingual, covering Python, C, and C++. Please refer to our response to W1.1 for the results on CWEval. We will clarify the programming-language coverage more explicitly in our revised paper.

---

> ### Author Response · Authors · 2025-11-25
> **Response to Reviewer WRYK (2/2)**
>
> > W2: Analysis of CWE types.
>
> We appreciate your suggestion to conduct a deep analysis of the CWE types. Following Chen et al. (ECOOP 2025),  we classified the tasks of CodeGuard+ into functionality-specific (Func) and non-functionality-specific (Non-Func). We then analyzed the improvement breakdown to determine the contribution of each category. As shown in the following table, we observe that LLMs generally perform better on non-functionality-specific vulnerabilities. Compared with zero-shot, RESCUE also achieves a larger performance gain in non-functionality-specific (+7.88) than functionality-specific vulnerabilities (+6.02) on average. We suppose this performance difference comes from the inherent nature of the benchmark tasks. Non-functionality-specific vulnerabilities in this benchmark typically involve checking explicit and local constraints (e.g., buffer size, null pointers). Such knowledge can be more easily summarized during knowledge extraction and incorporated into the generation process. By contrast, functionality-specific vulnerabilities (e.g., XXE, command injection, deserialization) arise from unsafe API usage and implicit threat models tied to particular libraries rather than violations of generic logic. These tasks require concrete domain-specific knowledge to prevent vulnerabilities. This may explain why zero-shot models struggle with them and why RESCUE achieves a smaller performance gain with functionality-specific CWE types.
>
> | Model | Type | \# Samples | Zero Shot | RESCUE | $\Delta$ |
> | :---- | :---- | :---- | :---- | :---- | :---- |
> | DeepSeek-Coder-V2-Lite | Func | 67 | 54.8 | 56.7 | \+1.9 |
> |  | Non-Func | 27 | 72.0 | 87.7 | \+15.7 |
> | Qwen2.5-Coder-7B | Func | 67 | 41.1 | 54.5 | \+13.4 |
> |  | Non-Func | 27 | 76.2 | 90.3 | \+14.1 |
> | Qwen2.5-Coder-32B | Func | 67 | 51.0 | 55.5 | \+4.5 |
> |  | Non-Func | 27 | 80.1 | 88.9 | \+8.8 |
> | Llama3.1-8B | Func | 67 | 48.4 | 49.3 | \+0.9 |
> |  | Non-Func | 27 | 66.9 | 73.3 | \+6.4 |
> | DeeSeek-V3 | Func | 67 | 53.6 | 63.0 | \+9.4 |
> |  | Non-Func | 27 | 91.9 | 86.3 | \-5.6 |

---

### Official Review · Reviewer_zcNL · 2025-10-30

**Soundness:** 3
**Presentation:** 3
**Contribution:** 3
**Rating:** 8
**Confidence:** 4

**Summary:**

The paper provides a framework called RESCUE, which aims to enhance LLMs to generate more secure code. The core innovations include a hybrid knowledge base construction method that uses LLM-assisted summarisation and static program slicing to distil raw security data into clear guidelines and concise code examples. Additionally, RESCUE employs a hierarchical multi-faceted retrieval system that proactively analyses coding tasks based on vulnerability causes, API patterns, and code similarity to accurately retrieve relevant security knowledge. Experiments across multiple benchmarks and six different LLMs demonstrate that RESCUE significantly improves both security and functional correctness. Ablation studies confirm the necessity of the specialised knowledge base construction and the effectiveness of the multi-faceted retrieval strategy.

**Strengths:**

- RESCUE consistently outperforms all existing methods across multiple benchmarks in terms of SecurePass@1, a comprehensive metric that jointly evaluates both functionality and security. This demonstrates the framework's ability to generate code that is not only correct but also resistant to common vulnerabilities, representing a significant advancement over prior approaches that often sacrifice one dimension for the other.
- The paper proposes a novel and systematic method for building a refined, hierarchical knowledge base from raw security data, effectively addressing the noise and redundancy present in conventional RAG designs. By combining LLM-assisted summarization with static program slicing techniques, RESCUE creates concise, actionable security guidelines and distilled code examples that are directly applicable to code generation tasks.

**Weaknesses:**

- The evaluation framework used in the paper, which relies on static security analysis tools, presents a key limitation: The evaluation framework employs static security analysis tools, which are known to potentially generate false positives and negatives
- RESCUE introduces some additional time cost to achieve its security improvements. Although the paper suggest the overhead is acceptable and can be further reduced through engineering optimizations, the additional costs are mostly unclear

**Questions:**

- What is the computational overhead?
- Have the authors checked for potential false positives in their static analysis?

---

> ### Author Response · Authors · 2025-11-24
> **Response to Reviewer zcNL**
>
> Dear Reviewer zcNL:
>
> We sincerely appreciate your constructive comments. We hope the following responses address your questions.
>
> > W1&Q2: The usage of static security analysis tools for evaluation.
>
> Thank you for raising this concern. We used static analysis tools in the evaluation to be consistent with  existing work on secure code generation \[1, 2, 3, 4\]. Indeed, static analysis tools are not perfect for evaluation. Yet it is still widely used in the literature since they are fully automated and there is no alternative evaluation method that scales to large benchmarks.
>
> During the rebuttal, we found a recent benchmark, CWEval \[5\], which overcomes the limitation of static analysis via dynamic testing. The authors of CWEval manually created test cases to assess the functional correctness and security of LLM-generated code. Due to the time limit, we have only compared RESCUE with zero-shot generation and SecCoder across three LLMs. The results are shown in the table below:
>
> | Model | Method | SecurePass@1 |
> | :---- | :---- | :---- |
> | Qwen2.5-Coder-7B | Zero Shot | 36.8 |
> |  | SecCoder | 36.0 |
> |  | RESCUE | **43.7** |
> | GPT-4o-mini | Zero Shot | 51.5 |
> |  | SecCoder | 52.44 |
> |  | RESCUE | **56.4** |
> | Deepseek-V3 | Zero Shot | 52.81 |
> |  | SecCoder | 61.91 |
> |  | RESCUE | **65.37** |
>
> The results show that RESCUE still consistently outperforms existing methods on a benchmark that uses a more robust evaluation method.
>
> \[1\] He, Jingxuan, Mark Vero, Gabriela Krasnopolska, and Martin Vechev. "Instruction Tuning for Secure Code Generation." In International Conference on Machine Learning, pp. 18043-18062. PMLR, 2024\.
>
> \[2\] Zhang, Boyu, Tianyu Du, Junkai Tong, Xuhong Zhang, Kingsum Chow, Sheng Cheng, Xun Wang, and Jianwei Yin. "SecCoder: Towards Generalizable and Robust Secure Code Generation." In Proceedings of the 2024 Conference on Empirical Methods in Natural Language Processing, pp. 14557-14571. 2024\.
>
> \[3\] He, Jingxuan, and Martin Vechev. "Large language models for code: Security hardening and adversarial testing." In Proceedings of the 2023 ACM SIGSAC Conference on Computer and Communications Security, pp. 1865-1879. 2023\.
>
> \[4\] Li, Dong, Meng Yan, Yaosheng Zhang, Zhongxin Liu, Chao Liu, Xiaohong Zhang, Ting Chen, and David Lo. "CoSec: On-the-Fly security hardening of code LLMs via supervised co-decoding." In Proceedings of the 33rd ACM SIGSOFT International Symposium on Software Testing and Analysis, pp. 1428-1439. 2024\.
>
> \[5\] Peng, Jinjun, Leyi Cui, Kele Huang, Junfeng Yang, and Baishakhi Ray. "Cweval: Outcome-driven evaluation on functionality and security of llm code generation." In 2025 IEEE/ACM International Workshop on Large Language Models for Code (LLM4Code), pp. 33-40. IEEE, 2025\.
>
> > W2&Q1: Overhead analysis of RESCUE.
>
> Thank you for this question. We provide a detailed analysis of the time cost in Appendix B.3. The table below reports the time cost of each step of RESCUE during inference (in seconds). The total inference time of RESCUE ranges from 10 seconds to 30 seconds depending on the underlying LLM. Furthermore, the majority of the additional overhead of the method comes from the Draft Code Generation and Vulnerability Cause Analysis steps, which require additional prompting to the LLM for code generation and analysis. Many coding agents these days already perform complex reasoning steps in code generation, which also introduces similar runtime overhead. So we think this additional time cost is reasonable and won’t lead to a significant slowdown in the developer workflow.
>
> There are several promising directions to mitigate this additional cost. For example, RESCUE can parallelize some LLM calls that do not have dependencies on each other to speed up the execution. We can also introduce a lightweight decision-maker at the beginning to selectively trigger RESCUE on incoming tasks that are security-relevant. If a task does not involve logic, function calls, or patterns associated with potential vulnerabilities, the coding agent can simply proceed with standard decoding without invoking RESCUE.
>
> | Model | Draft Code Generation | Vulnerability Cause Analysis | CWE-Level Retrieval | Code-Level Retrieval | Augmented Generation |
> | :---- | ----- | ----- | ----- | ----- | ----- |
> | Qwen2.5-Coder-7B | 2.8412 | 2.7365 | 0.0191 | 2.5892 | 3.2097 |
> | Llama3.1-8B | 3.5318 | 1.4546 | 0.0181 | 2.434 | 3.5409 |
> | DeepSeek-V3 | 12.7992 | 4.3321 | 0.0197 | 2.6314 | 10.9561 |

---

### Official Review · Reviewer_YZm1 · 2025-11-06

**Soundness:** 2
**Presentation:** 3
**Contribution:** 2
**Rating:** 6
**Confidence:** 3

**Summary:**

This paper proposes RESCUE, a RAG-based framework for secure code generation with two main innovations: (1) a hybrid knowledge base construction combining LLM-assisted cluster-then-summarize distillation with program slicing, and (2) hierarchical multi-faceted retrieval that analyzes API patterns, vulnerability causes, and code similarity. The method achieves 4.8% average improvement in SecurePass@1 across 4 benchmarks and 6 LLMs. The authors also performs ablations to find that the security knowledge base construction is helpful over directly security data and the hierarchical retrieval consistently outperforms non-hierarchical baseline.

**Strengths:**

- Core approach is technically sound with clear motivation
- Comprehensive experiments (4 benchmarks, 6 LLMs, 5 baselines)
- Ablation studies validating the main components
- The paper is generally well-organized structurally and is easy to understand

**Weaknesses:**

- The novelty of the contribution seems to be in applying the different components like cluster-and-summarize knowledge base with heirarchical retrieval. The method involves too many moving components like API pattern and vulnerability cause analysis. The gains based on the complexity of the system seems not much significant, which would limit the adoption of such an approach for secure code generation.
- The method involves many different hyperparameters like hop limit, thresholds for api and vulnerability analysis, batch size for cluster-and-summarize, etc. No justification is provided for how the value are chosen for most such hyperparameters.
- Given the complexity of the approach, it's possible that some of the gains in the paper come from overfitting

**Questions:**

- How were the threshold values (τ_API=4.0, τ_VA=0.75, τ_C=0.65) selected, and how sensitive are your results to these choices?
- Your training data contains only 9 CWE types for Python and 12 for C/C++, but the benchmark has 23 and 17 respectively (Table 8). Can you provide a breakdown of performance on seen vs. unseen CWE types, and explain how the method generalizes when no similar examples exist in the knowledge base?
- Given that RESCUE seems to adds 3-4x computational overhead compared to zero-shot generation, can you provide a cost-benefit analysis showing when this overhead is justified, and propose a lightweight filtering mechanism to apply RESCUE only to security-critical code? This could be big enough to be part of the future work.
-

---

> ### Author Response · Authors · 2025-11-24
> **Response to Reviewer YZm1 (1/n)**
>
> Dear Reviewer YZm1,
>
> Thank you very much for your comments\! We hope our following response can answer your questions.
>
> > W1: Complexity about RESCUE
>
> Thank you for raising this point. In this work, we address several common limitations of existing RAG methods for secure code generation, including data noise and redundancy in the knowledge base and the limitation of treating security knowledge as general text during retrieval. Thus, we need to come up with several components to address these limitations. While this makes our method look more complex than other RAG methods, our ablation study in Section 3.3 demonstrates that each individual component is necessary and makes a meaningful contribution to the final performance of RESCUE (Table 2, Table 3, and Figure 3). In particular, these components collectively lead to a significant gain compared with other RAG-based secure coding generation methods. For example, compared with SecCoder \[1\], RESCUE improves SP@1 by 7.5 absolute points.
>
> More importantly, real-world security vulnerabilities often result in substantial monetary losses in production. For instance, the average cost of a data breach is estimated to reach 4.4 million USD in 2025 \[2\]. Unlike functional bugs, security vulnerabilities typically have far broader and more severe consequences. This means the improvement of security can contribute to more practical benefits. Moreover, following the “Rule of 100” from IBM, solving a bug earlier can reduce the associated cost by 100x \[3\]. Therefore, adoption of RESCUE at the development stage would be a worthwhile investment.
>
> We also believe our detailed analysis of individual components in RESCUE can provide empirical insights into the design space of RAG-based secure code generation. By breaking down and evaluating each part of the system, we provide evidence that can inform different design decisions. Researchers and practitioners can then selectively adopt the components that best align with their needs, preferences, and constraints, rather than treating RESCUE as an all-or-nothing solution. Furthermore, the knowledge construction process is a one-time effort and the resulting security guidelines and sliced code examples can be used by other RAG systems as well.
>
> \[1\] Zhang, Boyu, Tianyu Du, Junkai Tong, Xuhong Zhang, Kingsum Chow, Sheng Cheng, Xun Wang, and Jianwei Yin. "SecCoder: Towards Generalizable and Robust Secure Code Generation." In Proceedings of the 2024 Conference on Empirical Methods in Natural Language Processing, pp. 14557-14571. 2024\.
>
> \[2\] [https://www.ibm.com/reports/data-breach](https://www.ibm.com/reports/data-breach)
>
> \[3\] https://cloudqa.io/how-much-do-software-bugs-cost-2025-report/

---

> > ### Author Response · Authors · 2025-11-24
> > **Response to Reviewer YZm1 (n/n)**
> >
> > > W3: Overfitting
> >
> > Regarding the concern about overfitting, our method has been evaluated on six large language models and four benchmarks. This comprehensive evaluation demonstrates that the performance of RESCUE is consistent and generalizable across different scenarios. During the rebuttal period, we have also included an additional benchmark called CWEval \[1\]. This benchmark uses dynamic testing to rigorously evaluate the functional correctness and security through test cases. Due to the time limit, we have only compared RESCUE with zero-shot generation and SecCoder across three LLMs. As shown in the following table, RESCUE consistently outperforms zero-shot generation and SecCoder.
> >
> > | Model | Method | SecurePass@1 |
> > | :---- | :---- | :---- |
> > | Qwen2.5-Coder-7B | Zero Shot | 36.8 |
> > |  | SecCoder | 36.0 |
> > |  | RESCUE | **43.7** |
> > | GPT-4o-mini | Zero Shot | 51.5 |
> > |  | SecCoder | 52.44 |
> > |  | RESCUE | **56.4** |
> > | Deepseek-V3 | Zero Shot | 52.81 |
> > |  | SecCoder | 61.91 |
> > |  | RESCUE | **65.37** |
> >
> > \[1\] Peng, Jinjun, Leyi Cui, Kele Huang, Junfeng Yang, and Baishakhi Ray. "Cweval: Outcome-driven evaluation on functionality and security of llm code generation." In 2025 IEEE/ACM International Workshop on Large Language Models for Code (LLM4Code), pp. 33-40. IEEE, 2025\.
> >
> > > Q2: Breakdown and analysis of performance on seen and unseen CWE types.
> >
> > Thank you for asking. During the rebuttal, we did a more fine-grained analysis on the CodeGuard+ benchmark based on CWE types present in our knowledge base ("seen") and those that are not ("unseen"). The table below shows the performance of RESCUE on seen and unseen CWEs.
> >
> > | Model | Type | Count | Zero Shot | RESCUE | $\Delta$ |
> > | :---- | :---- | :---- | :---- | :---- | :---- |
> > | DeepSeek-Coder-V2-Lite | seen | 52 | 68.2 | 78.1 | \+9.9 |
> > |  | unseen | 42 | 49.3 | 50.2 | \+0.9 |
> > | Qwen2.5-Coder-7B | seen | 52 | 56.6 | 76.8 | \+20.2 |
> > |  | unseen | 42 | 44.5 | 49.9 | \+5.4 |
> > | Qwen2.5-Coder-32B | seen | 52 | 64.6 | 73.3 | \+8.7 |
> > |  | unseen | 42 | 52.8 | 55.0 | \+2.2 |
> > | Llama3.1-8B | seen | 52 | 65.2 | 64.2 | \-1.0 |
> > |  | unseen | 42 | 39.5 | 46.2 | \+6.7 |
> > | DeeSeek-V3 | seen | 52 | 71.1 | 76.9 | \+5.8 |
> > |  | unseen | 42 | 56.4 | 60.7 | \+4.3 |
> >
> > According to the table, even though RESCUE does not achieve as much improvement on unseen CWE types, RESCUE can still help address some unseen CWE types, leading to an absolute improvement of 3.9 on average. We suppose this generalizability comes from several factors:
> >
> > 1. Different CWE types, such as different types of buffer overflows, could share similar vulnerability patterns.
> > 2. Security guidelines are common and universal even for different CWE scenarios.
> > 3. LLMs are trained on massive code data, meaning that the security context can activate relevant parametric knowledge in LLMs.
> >
> > Finally, we’d like to note that the offline knowledge construction process of RESCUE is fully automated and can be applied to construct a substantially larger knowledge base covering more CWE types. In our experiments, we constructed a knowledge base with only 18 CWE types because the available security dataset we used as the information source was relatively small. The method itself, however, is not limited by this choice and can scale to broader vulnerability sets.
> >
> > > Q3: Overhead analysis
> >
> > We agree with the reviewer that RESCUE indeed introduces additional cost compared with zero-shot generation. We believe such a cost is worthwhile given the security improvement, e.g., an average of 6.3% absolute improvement in SecurePass@1. As mentioned in our response to W1, the consequence of introducing a security vulnerability far exceeds the additional overhead in our method. Furthermore, based on our cost analysis in Appendix B.3, the total inference time of RESCUE ranges from 10 seconds to 30 seconds depending on the underlying LLM. Given that many coding agents already perform complex reasoning steps in code generation, which also introduce similar runtime overhead, we believe the overhead of RESCUE is acceptable to developers and won’t lead to a significant slowdown in the developer workflow.
> >
> > We also agree with your suggestion that a lightweight filtering mechanism is a promising direction for future work, as not all tasks require security-focused processing. Since RESCUE already involves proactive multi-faceted analysis for security, this step can be leveraged to analyze the security relevance of an incoming task. If the task does not involve logic, function calls, or patterns associated with potential vulnerabilities, the coding agent can simply proceed with standard decoding without invoking RESCUE.
> > Alternatively, we can also fine-tune a specific small model to analyze security intents or requirements for an incoming task, thereby reducing the cost of invoking large models.

---

> ### Author Response · Authors · 2025-11-24
> **Response to Reviewer YZm1 (2/n)**
>
> > W2 & Q1: Hyperparameter justification and sensitivity analysis about thresholds.
>
> Thank you for this question. The 2-hop limit was chosen as a trade-off to capture the necessary context while avoiding unnecessarily long code examples, whereas the batch size of 10 was determined by the context window constraints of the summarization model. Regarding retrieval thresholds, our primary target is to filter out low-relevance retrieved results, which receive low scores and may be noise for generation. Thus, the exact threshold is relatively insensitive and remains stable across a reasonable range. To support this, we conducted a sensitivity analysis. Due to time and computation constraints, we randomly sampled 20 instances from CodeGuard+ as validation data and tested Llama3.1-8B-Instruct and Qwen2.5-Coder-7B-Instruct. By fixing two thresholds and varying the third, the following tables show the results, where numbers in parentheses denote the change relative to zero-shot. We observed that the overall trend remains stable, confirming the robustness of threshold selection. We will include these justifications and the sensitivity analysis results in the revised paper.
>
> For API Threshold:
>
> | Model | API Threshold | SecurePass@1 | Pass@1 | SecureRate |
> | :---- | ----- | ----- | ----- | ----- |
> | Llama3.1-8B-Instruct | 4 (default) | 56.7 (+8.4) | 86.7 (-1.6) | 66.7 (+11.7) |
> |  | 0.0 | 51.7 (+3.4) | 81.7 (-6.6) | 65.0 (+10.0) |
> |  | 2.0 | 53.3 (+5.0) | 86.7 (-1.6) | 65.0 (+10.0) |
> |  | 3.0 | 53.3 (+5.0) | 78.3 (-10.0) | 66.7 (+11.7) |
> |  | 5.0 | 51.7 (+3.4) | 83.3 (-5.0) | 65.0 (+10.0) |
> |  | 7.0 | 50.0 (+1.7) | 80.0 (-8.3) | 65.0 (+10.0) |
> | Qwen2.5-Coder-7B | 4 (default) | 61.7 (+5.0) | 91.7 (-1.6) | 65.0 (+6.7) |
> |  | 0 | 59.5 (+2.8) | 94.5 (+1.2) | 63.7 (+5.4) |
> |  | 2 | 61.0 (+4.3) | 94.5 (+1.2) | 62.9 (+4.6) |
> |  | 3 | 58.5 (+1.8) | 92.0 (-1.3) | 61.8 (+3.5) |
> |  | 5 | 62.0 (+5.3) | 94.5 (+1.2) | 64.5 (+6.2) |
> |  | 7 | 62.5 (+5.8) | 94.5 (+1.2) | 64.2 (+5.9) |
>
> For Vulnerability Cause Analysis Threshold:
>
> | Model | Vulnerability Cause Analysis Threshold | SecurePass@1 | Pass@1 | SecureRate |
> | :---- | ----- | ----- | ----- | ----- |
> | Llama3.1-8B-Instruct | 0.75(default) | 56.7 (+8.4) | 86.7 (-1.6) | 66.7 (+11.7) |
> |  | 0.00 | 46.7 (-1.6) | 80.0 (-8.3) | 61.7 (+6.7) |
> |  | 0.25 | 50.0 (+1.7) | 81.7 (-6.6) | 63.3 (+8.3) |
> |  | 0.45 | 46.7 (-1.6) | 76.7 (-11.6) | 63.3 (+8.3) |
> |  | 0.65 | 55.0 (+6.7) | 88.3 (+0.0) | 61.7 (+6.7) |
> |  | 0.85 | 45.0 (-3.3) | 75.0 (-13.3) | 58.3 (+3.3) |
> |  | 0.95 | 51.7 (+3.4) | 86.7 (-1.6) | 60.0 (+5.0) |
> | Qwen2.5-Coder-7B | 0.75(default) | 61.7 (+5.0) | 91.7 (-1.6) | 65.0 (+6.7) |
> |  | 0.00 | 58.5 (+1.8) | 90.5 (-2.8) | 64.5 (+6.2) |
> |  | 0.25 | 62.0 (+5.3) | 92.5 (-0.8) | 64.4 (+6.1) |
> |  | 0.45 | 60.0 (+3.3) | 93.5 (+0.2) | 61.0 (+2.7) |
> |  | 0.65 | 61.5 (+4.8) | 95.0 (+1.7) | 64.0 (+5.7) |
> |  | 0.85 | 60.0 (+3.3) | 95.5 (+2.2) | 62.4 (+4.1) |
> |  | 0.95 | 58.0 (+1.3) | 93.5 (+0.2) | 61.8 (+3.5) |
>
> For Code Threshold:
>
> | Model | Code Threshold | SecurePass@1 | Pass@1 | SecureRate |
> | :---- | ----- | ----- | ----- | ----- |
> | Llama3.1-8B-Instruct | 0.65 (default) | 56.7 (+8.4) | 86.7 (-1.6) | 66.7 (+11.7) |
> |  | 0.00 | 55.0 (+6.7) | 80.0 (-8.3) | 65.0 (+10.0) |
> |  | 0.15 | 55.0 (+6.7) | 83.3 (-5.0) | 65.0 (+10.0) |
> |  | 0.35 | 50.0 (+1.7) | 81.7 (-6.6) | 66.7 (+11.7) |
> |  | 0.55 | 51.7 (+3.4) | 80.0 (-8.3) | 65.0 (+10.0) |
> |  | 0.75 | 46.7 (-1.6) | 70.0 (-18.3) | 66.7 (+11.7) |
> |  | 0.85 | 45.0 (-3.3) | 75.0 (-13.3) | 63.3 (+8.3) |
> |  | 0.95 | 46.7 (-1.6) | 75.0 (-13.3) | 65.0 (+10.0) |
> | Qwen2.5-Coder-7B | 0.65 (default) | 61.7 (+5.0) | 91.7 (-1.6) | 65.0 (+6.7) |
> |  | 0.00 | 59.5 (+2.8) | 92.0 (-1.3) | 60.4 (+2.1) |
> |  | 0.15 | 62.0 (+5.3) | 95.5 (+2.2) | 62.7 (+4.4) |
> |  | 0.35 | 61.0 (+4.3) | 97.5 (+4.2) | 61.6 (+3.3) |
> |  | 0.55 | 61.0 (+4.3) | 93.5 (+0.2) | 61.7 (+3.4) |
> |  | 0.75 | 60.0 (+3.3) | 92.5 (-0.8) | 60.0 (+1.7) |
> |  | 0.85 | 60.0 (+3.3) | 93.0 (-0.3) | 61.1 (+2.8) |
> |  | 0.95 | 61.5 (+4.8) | 94.5 (+1.2) | 61.8 (+3.5) |

---

### Author Response · Authors · 2025-12-04
**Final Summary of Discussion**

Dear Reviewers and ACs:

We sincerely appreciate your additional efforts in reviewing and evaluating our work under the new ICLR policy. Below, we summarize the key discussion outcomes.

We are encouraged that our work has been recognized for:
1\. A timely and important research problem (Reviewers WRYK, M2w4).
2\. A well-motivated method with a novel, sound, and comprehensive design (Reviewers YZm1, zcNL, WRYK).
3\. Strong performance through extensive experiments and ablation studies (Reviewers YZm1, zcNL, WRYK, M2w4).
4\. This paper is clearly written and well-organized (Reviewers YZm1, M2w4).

During the discussion period, we conducted additional experiments, analyses, and clarification to address the reviewers' questions:
1\. We added CWEval, a new security benchmark which uses dynamic testing to jointly evaluate security and functional correctness. The results further validate RESCUE's effectiveness.
2\. We provided detailed clarification of the difference between our work and recent related work, highlighting our technical novelties and contributions.
3.We broke down the results into seen vs. unseen CWE types and evaluated our method on additional programming languages (Go and JavaScript), demonstrating effective generalization to new scenarios.
4\. We also included a deeper analysis of functionality-specific vs. non-functionality-specific vulnerabilities.
5\. We performed additional ablation studies on our modified RRF design and a sensitivity analysis of its hyperparameters, confirming both effectiveness and robustness.
6\. We provided qualitative and quantitative overhead analyses, justifying that the cost is reasonable and acceptable given the significant security improvements.

Best regards,
Authors

---

### Meta-Review · Area_Chair_9LGa · 2025-12-31

**Summary:**

This paper proposes RESCUE, a retrieval-augmented framework for secure code generation that combines (1) a hybrid knowledge base construction strategy integrating LLM-assisted cluster-and-summarize distillation with static program slicing, and (2) a hierarchical, multi-faceted retrieval mechanism that reasons over vulnerability causes, API usage patterns, and code similarity.

Reviewers generally agree that the problem is timely and important, the method is well-motivated and technically sound, and the experimental evaluation is extensive with meaningful ablations.
While some reviewers raised concerns regarding system complexity, evaluation realism, and computational overhead, the rebuttal substantially strengthened the paper by adding new benchmarks, deeper analyses, and clearer justification of design choices.

**Reviewer Concerns:**

Concerns largely addressed by the rebuttal:

- Evaluation validity and realism:
Reviewers questioned the reliance on static analysis and programming-contest-style benchmarks. The authors addressed this by adding results on CWEval, a dynamic-testing-based benchmark, and showed consistent improvements over zero-shot and SecCoder. This significantly strengthens confidence in the reported gains.

- Generalization and overfitting:
Concerns about limited CWE coverage and potential overfitting were addressed through a detailed breakdown of seen vs. unseen CWE types, multilingual evaluation (Python, C/C++, Go, JavaScript), and consistent performance across six LLMs.

- Hyperparameter sensitivity:
The authors provided explicit justification and sensitivity analyses for key thresholds, demonstrating that performance is relatively stable across a reasonable range of values.

- Lack of vulnerability-type analysis:
The rebuttal added a detailed analysis distinguishing functionality-specific vs. non-functionality-specific vulnerabilities, clarifying how different CWE categories contribute to the overall improvements.

- Computational overhead:
The authors provided a detailed cost breakdown, showing end-to-end inference times of 10–30 seconds depending on the LLM, and argued convincingly that this overhead is acceptable in security-critical contexts. They also outlined concrete strategies for selective or lightweight deployment.

Remaining or partially addressed concerns:

- System complexity and adoption cost:
Some reviewers remain concerned that RESCUE involves many components and may be complex to deploy in practice. While the authors argue that security gains justify the cost and that components can be adopted selectively, this remains a trade-off rather than a flaw.

- Scope of real-world deployment:
Although CodeGuard+ and CWEval mitigate earlier concerns, further large-scale, repo-level deployment studies could strengthen future work.

**Reviewer Scores:**

Based on the discussion and rebuttal, I believe most reviewers would maintain or slightly increase their scores:

- Reviewers who initially gave borderline accept scores (e.g., 6) would likely remain positive given the added benchmarks and analyses.

- Stronger reviewers would likely maintain their acceptance recommendation, as their main questions were directly addressed.

---

### Decision · Program_Chairs · 2026-01-26

Accept (Poster)